# Use of Algerian Type *Ras El-Hanout* Spices Mixture with Marination to Increase the Sensorial Quality, Shelf Life, and Safety of Whole Rabbit Carcasses under Low-O_2_ Modified Atmosphere Packaging

**DOI:** 10.3390/foods12152931

**Published:** 2023-08-02

**Authors:** Djamel Djenane, Yamina Ben Miri, Agustín Ariño

**Affiliations:** 1Food Quality and Food Safety Research Laboratory, Department of Food Sciences, Mouloud Mammeri University, P.O. Box 17, Tizi Ouzou 15000, Algeria; sys_yamina@yahoo.com; 2Department of Biochemistry and Microbiology, Faculty of Sciences, Mohamed Boudiaf University, P.O. Box 166, M’sila 28000, Algeria; 3Facultad de Veterinaria, Instituto Agroalimentario de Aragón-IA2, Universidad de Zaragoza-CITA, 50013 Zaragoza, Spain; aarino@unizar.es

**Keywords:** rabbit meat, Algerian *Ras El-Hanout*, marinating solution, retail display, low-O_2_ MAP, quality, shelf life, safety

## Abstract

This study aimed to evaluate the effect of combined treatments with *Ras El-Hanout* spices mixture and marinade solution containing extra virgin olive oil, onion, garlic, and concentrated lemon juice on sensorial quality, shelf life, and safety of whole rabbit carcasses under low-O_2_ modified atmosphere packaging (MAP). The values of pH, water holding capacity, shear force, thiobarbituric acid reactive substances, total volatile basic nitrogen, color (CIE L*a*b*), sensorial tests, and spoilage microorganisms were determined in rabbit meat at 0, 5, 10, 15, and 20 days during a retail display at 7 ± 1 °C. The results indicated that the marination process using the *Ras El-Hanout* blend of spices improved the water-holding capacity of meat maintaining optimum pH values. This combined treatment delayed the growth of major spoilage microorganisms, lipid oxidation, protein degradation, and undesirable color changes compared to unmarinated samples from the fifth to the twentieth day of retail exposure. The shelf life of rabbit carcasses under low-O_2_ MAP could be extended to 20 days of retail display, while rabbit carcasses under aerobic display presented a shorter shelf life of 5 to 10 days. Instrumental and sensorial tests showed that low-O_2_ MAP enhanced the tenderness of whole rabbit carcasses, with those marinated with *Ras El-Hanout* being the most positively perceived by the panelists. Marination also inhibited the pathogen *Campylobacter jejuni*, thus increasing the microbiological safety of the packaged product. The overall results indicated that low-O_2_ MAP combined with the *Ras El-Hanout* spice blend and marinade solution may represent a promising strategy for retail establishments to improve the quality, shelf life, and safety of rabbit carcasses.

## 1. Introduction

In Algeria, a traditional rabbit-producing country, the rabbit sector has not experienced the expected economic prosperity, unlike other meat sectors such as poultry, beef, sheep, and goat. Current rabbit meat consumption is low (<0.4 kg/capita/year), due to the lack of meat supply and the unavailability of marketing. According to the Food and Agriculture Organization of the United Nations statistical database [1], the total production of rabbit meat worldwide reached approx. 860,000 tons in the 2021 year, and Algeria ranked among the seven largest rabbit meat-producing countries with a production of 8474 tons. FAOSTAT listed China, North Korea, and Egypt as the world’s leading producers of rabbit meat.

The changes in consumer lifestyles in developed countries during the last few years have led to a meat market more addressed towards convenience food, which is “easy-handled and processed products”. This trend has been exploited for a long time by meat industries, and more newly by the rabbit sector too. In Mediterranean regions, there has been an increased interest by consumers in healthy meats. Rabbit meat can be considered a functional food that offers bioactive compounds with favorable effects on human health, such as conjugated linoleic acid (CLA), as well as a well-adjusted ratio of n-6 to n-3 polyunsaturated fatty acids (PUFAs). Rabbit meat contains low quantities of fat, cholesterol, and sodium (Na) and high content of PUFAs, potassium (K), magnesium (Mg), and phosphorus (P); for these reasons, it is recommended mostly for people with cardiovascular illnesses and children [2].

In Algeria, most rabbit meat is still sold as fresh whole carcasses without any kind of protection, so processed products are needed because of their suitability and high quality and safety standards, which will stimulate the rabbit industry to adopt better procedures for the production of new processed meats. Improving the quality of rabbit meat is vital to motivate its consumption.

Meat packaging is rapidly moving from simple protective materials to optimizing meat cuts, designing ready-to-eat foods, extending shelf life, and ensuring safety and authenticity. Current trends in meat packaging have shifted from conventional vacuum packaging to modified atmosphere packaging (MAP), active packaging, and nanotechnology [3,4,5]. Therefore, emerging methods that are low-cost, effective, and safe to increase the stability of fresh rabbit meat at retail are of great importance to the rabbit industry. Synthetic preservatives have been used in meat industries to prevent spoilage and improve safety. Recently, growing consumer awareness of natural bioactive compounds is palpable, as these compounds could be valid alternatives to partially replace synthetic compounds and avoid harmful side effects on human health [6]. Moreover, due to their beneficial effects on digestive function and growth, and their antimicrobial and antioxidant properties, the use of spices and herbs containing bioactive molecules, such as polyphenols, as feed supplements in animal production has been investigated in the last decade to improve rabbit health and meat quality attributes [7,8].

The current demand for marinated meat products has increased significantly. The implications of this scenario are associated above all with nutritional characteristics, prolongation of shelf life, and improvement of the sensory and textural characteristics of this type of product [9,10,11]. In addition, marinating technology makes it possible to expand the range of meat products and, by conferring them with peculiar sensory characteristics, offer consumers greater variety. Several spice mixes are used in gastronomy, to combine taste properties and produce antioxidant and antimicrobial actions [12]. *Ras El-Hanout* (translated as “head of the shop”) is a blend of spices known for its ancestral popularity in North Africa, which is used especially in traditional dishes in the Maghreb region [13].

Among bacteria associated with foodborne diseases, *Campylobacter* has been the most frequently isolated pathogen in outbreaks in both developed and developing countries in the last 10 years. Thus, *Campylobacter* infection is one of the most common causes of bacterial gastroenteritis in humans and is often transmitted asymptomatically in animal reservoirs [14,15]. *Campylobacter* spp. have relatively demanding growth requirements as most isolates require microaerophilic conditions (5% O_2_ and 10% CO_2_) to grow. Handling and processing of rabbit carcasses in the home and in public places, such as retail sales, are also important sources of contamination. Worse, mixed farming of chickens and rabbits could increase the risk of pathogen transmission.

To our knowledge, information on the effects of marinade combined with *Ras El-Hanout* spice mix, ingredients, and seasonings typical of Algerian gastronomy, on the quality and safety of rabbit carcasses is scarce. Therefore, the purpose of the present work was to evaluate the effect of the addition of a marinade solution plus *Ras-El-Hanout* spice blend, on the sensory, quality, shelf life, and safety parameters of whole rabbit carcasses packed in low-O_2_ MAP during retail exposure at 7 ± 1 °C for 20 days. The opportunity to improve the organoleptic and microbiological quality of rabbit meat with bioactive compounds represents a sustainable strategy that could revive the rabbit meat sector and could certainly offer added value to the final product and differentiate it from the alternatives currently on the market.

## 2. Materials and Methods

### 2.1. Ingredients of Spices Mixture and Marinade Solution

The marinade solution was composed of the following ingredients: extra virgin olive oil (EVOO), garlic, onion, and concentrated lemon juice. *Ras El-Hanout* consists of a blend of spices and aromatic herbs used throughout North Africa and the Middle East and is mainly associated with Maghrebi cuisine. There is no single recipe for preparing the *Ras El-Hanout* mixture. In fact, the composition varies from one region to another within each of the Maghreb countries, depending on the geographical and agricultural conditions and the customs and habits of each region. The *Ras El-Hanout* blend used in this study was purchased at a supermarket in Tizi-Ouzou (Algeria) and contained clove (*Syzygium aromaticum*), thyme (*Thymus vulgaris*), oregano (*Origanum vulgare*) cinnamon (*Cinnamomum verum*), cumin (*Cuminum cyminum*), black pepper (*Piper nigrum*), sweet red pepper (*Capsicum annuum*), coriander (*Coriandrum sativum*) and turmeric (*Curcuma longa*).

The marinade solution is prepared with EVOO, garlic, onion, and concentrated lemon juice mixed with *Ras El-Hanout*—a mixture of typical ingredients species: clove, thyme, oregano, cinnamon, cumin, black pepper, sweet red pepper, coriander, and turmeric—used in our work was designated because they offer a marinated product within the Mediterranean diet. All the marinade ingredients come from natural plants commonly used in the Algerian area and constitute an added value to the product itself.

### 2.2. Optimization of the Spices Mixture and Marinade Solution

Generally, the addition of marinade solutions to rabbit meat is carried out to improve yield by increasing the moisture content of the product. Marinating consists of submerging the meat matrix in aqueous solutions containing a wide range of ingredients, such as essential oils (EO), vinegar, water, lemon juice, salt, tenderizers, herbs, wine, beer, organic acids, and spices. Depending on the ingredients selected, there is a wide variety of marinade solutions, alkaline or acidic. The choice of the basic compounds (EVOO, garlic, onion, and lemon) and *Ras El-Hanout* was determined according to the gastronomic traditions of the Algerian area. Preliminary tests were carried out to select the most suitable combination. The marinade solution compositions considered during these tests were EVOO, lemon, garlic, and onion as basic compounds and various proportions of *Ras El-Hanout* mixture between 200 and 900 mg/kg in the final marinade solution. Preliminary tests were conducted considering the organoleptic parameters of rabbit meat, such as odor, color, and overall acceptability before and after cooking. Based on the preliminary results, the final marinade solution selected for the main experiment was composed of 4 parts EVOO, 2 parts lemon, 1 part garlic, and 1 part onion (4/2/1/1, In other words: 50/25/12.5/12.5%) with a *Ras El-Hanout* mixture at 500 mg/kg (0.05%). To obtain homogeneous marinade solutions throughout the study, the ingredients were mixed in Microtron^®^ MB800 (Kinematica AG, Malters, Switzerland) at 12,000 rpm for 1 min.

Total phenolic compounds (TPCs) in the final marinade solution were determined spectrophotometrically according to the Folin–Ciocalteu method with some modifications [16]. Gallic acid (GA) was used as standard phenolic compound for the calibration curve. Results were expressed as milligrams of gallic acid equivalents (GAE) per gram of sample dry weight (mg GAE/g).

### 2.3. Campylobacter jejuni Strain and Standardization of Inoculum

The foodborne pathogen used in this work for the challenge trial was *Campylobacter jejuni* subsp. *jejuni*. To standardize the number of cells, *C. jejuni* was grown in Bolton broth (CM0983, Oxoid Ltd., Hampshire, UK) supplemented with 5% lysed horse blood (SR0048) and selective supplement (SR0183) at 42 °C in microaerophilic conditions (5% O_2_, 10% CO_2_, and 85% N_2_). An amount of 1 mL of this culture was standardized through two successive 24 h growth cycles at 42 °C in 9 mL of brain–heart infusion (BHI) broth (CM1135). After successive growth, 50 mL of the suspension was then inoculated in fresh BHI broth and incubated at 42 °C for 12 h. The final bacterial load was approximately 3 × 10^8^ CFU/mL, determined by measuring transmittance at 600 nm (Spectronic 20, Bausch & Lomb, Rochester, NY, USA). Count was made on *Campylobacter* blood-free selective medium (CCDA) (CM0739 plus selective supplement SR0155) under microaerophilic conditions by seeding 1 mL of the dilutions −5, −6, and −7 to verify the veracity of the spectrophotometer readings. The *C. jejuni* strain was maintained frozen (−80 °C) in cryovials (Cryobanks, Mast, Merseyside-Liverpool, UK) containing an antifreezing agent to preserve the viability of the cells during storage and was subcultured for each assay.

### 2.4. Rabbit Carcasses Treatments

Samples of fresh rabbit carcasses were acquired from an establishment located in the city of Aïn El-Hamra (Bordj-Ménaïel), Algeria, consisting of mixed rabbit and broiler breeding facilities in open-air parks. Sixty-four healthy male rabbits (white breed) were slaughtered following the *Halal* procedure and whole carcasses with an average weight of 1.28 kg were obtained (Animal Welfare: The rabbits used for this trial were cared for in accordance with the guidelines from the Algerian Ministry of Agriculture (Arrêtés 1 August 1984 and 15 July 1996). Carcasses were aseptically transported cold (1 ± 1 °C) to the Food Safety laboratory of the University of Tizi-Ouzou (Algeria) within 2 h. Carcasses were kept strictly at 1 ± 1 °C and all subsequent preparations were performed at room temperature (20 °C). On the day after slaughter (6 h *post mortem*), eviscerated rabbit carcasses were individually weighed and assigned to one of the two groups. The 64 rabbit carcasses were divided equally into 2 groups (Group 1 and 2) in turn subdivided into 4 batches with 8 carcasses each as follows:Group 1: Unmarinated rabbit carcasses (n = 32)1-A: Packed in aerobic conditions.1-MA: Packed in low-O_2_ modified atmosphere.1-CA: Contaminated with *C. jejuni* and packed in aerobic conditions.1-CMA: Contaminated with *C. jejuni* and packed in low-O_2_ modified atmosphere.Group 2: Marinated rabbit carcasses (n = 32)2-A: Packed in aerobic conditions.2-MA: Packed in low-O_2_ modified atmosphere.2-CA: Contaminated with *C. jejuni* and packed in aerobic conditions.2-CMA: Contaminated with *C. jejuni* and packed in low-O_2_ modified atmosphere.

Each rabbit carcass was aseptically processed, individually packaged into a polystyrene tray, and completely enveloped with transparent polyethylene/polyamide (PE/PA) coextruded film supplied by the Department of Animal Production and Food Science of the University of Zaragoza (Spain). For marinating, the marinade solution with *Ras El-Hanout* was added to the carcasses at a level corresponding to 8% (*v*/*w*) of the final product. To ensure a better marinating operation, treated carcass samples were further blended for 1 min. For the packaging, the trays were filled with air or with a microaerophilic gas mixture of 5% O_2_ + 40% CO_2_ + 55% N_2_. For the challenge trial, carcasses were placed into sterile plastic bags and inoculated with *C. jejuni* pathogen (2 × 10^5^ CFU/g). In order to ensure a good distribution of pathogenic microorganisms over the whole surface rabbit carcasses, the latter was manually stirred for 1 min, which represents the time needed to obtain a good bacterial surface distribution. All samples were stored at 7 ± 1 °C to represent retail display temperature conditions currently applied in Algerian fresh meat markets. For each selected sampling time (0, 5, 15, and 20 days), two trays of each batch, each containing a carcass were randomly selected, and their hind leg (HLs) portions were aseptically excised, and visible subcutaneous fat and connective tissue were removed (Figure 1). Eight of them (uncontaminated) were used for pH determination and analysis of sensorial parameters (color, Warner–Bratzler shear force, water holding capacity, cooking losses, sensory panel), thiobarbituric acid-reactive substances, total volatile basic nitrogen, and spoilage microorganisms (plate count agar, *Pseudomonas* spp., and *Brochothrix thermosphacta*), and the other eight were used exclusively for foodborne pathogen analysis (*C. jejuni*). All analyses were run in triplicate.

### 2.5. Determination of pH of Rabbit Meat

The pH of the meat was measured using a portable pH-METER (Hanna instruments, HI2002-02, Lingolsheim, France). The pH of the resultant homogenates (diluted 1:10 in distilled water) was measured using the electrode attached to the pH meter.

### 2.6. Color Measurements in Surface Rabbit Meat

Rabbit meat color was measured at room temperature on the carcass’s surface using a portable colorimeter Chroma Meter CR-400 Konica Minolta Sensing (Minolta Sensing Inc., Osaka, Japan). For color measurements, since marinated carcasses show a non-uniform coloration, color was evaluated over the entire surface of the legs. Color measurements were reported in terms of lightness (L*), redness (a*), and yellowness (b*) in the CIE Lab color space model [17]. The color values were obtained considering the average of ten (10) readings per sample.

### 2.7. Warner–Bratzler Shear Force of Rabbit Meat

Instrumental tenderness was measured using the Warner–Bratzler (WB) method as described by Djenane et al. [3]. Therefore, to avoid excessive variability between samples, it is important to consider the direction in which the muscle is cut in relation to the orientation of the muscle fibers. At least six cores (1 cm diameters × 3 cm in length) from each rabbit leg portion were removed parallel to the longitudinal orientation of the muscle fibers. The cores were sheared perpendicular to the muscle fibers’ orientation using an Alliance RT/5 (MTS Systems Corp., Eden Prairie, MN, USA) with a WB shear device and crosshead speed set at 2 mm/s. Results were expressed as load in kg.

### 2.8. Water-Holding Capacity and Cooking Losses of Rabbit Meat

Water-holding capacity (WHC) expressed as released water (RW%) was carried out by a method of compression. WHC was assessed on a rabbit meat sample (3 g), placed on desiccated filter paper (10 cm diameter). The rabbit meat sample and the filter paper were placed between two Plexiglas sheets and immediately pressed for 5 min by a 2.25 kg weight. After that, the moist filter paper was fast-weighed after removal of the compressed meat. The WHC expressed as weight of released water (%) was calculated by the following equation:(1)$$\mathrm{RW}\;(\%)=\frac{\left(\mathrm{weight}\;\mathrm{of}\;\mathrm{moist}\;\mathrm{filter}\;\mathrm{paper}\;-\;\mathrm{weight}\;\mathrm{of}\;\mathrm{dessicated}\;\mathrm{filter}\;\mathrm{paper}\right)}{\left(\mathrm{initial}\;\mathrm{weigh}\;\mathrm{of}\;\mathrm{meat}\right)}\times 100$$

For cooking losses (%), rabbit meat cuts from carcasses (5 × 5 × 2 cm) were weighed and then individually wrapped in heat-resistant polyethylene bags, labeled, and heat-treated in a 150 °C oven to an internal temperature of ~70 °C. The cuts were turned every 2 min to prevent excess surface crust formation. After cooking, samples were removed and cooled to room temperature (20 °C) under tap water for 20 min [18]. Cooking loss (%) was calculated as follows:(2)$$\mathrm{Cooking}\;\mathrm{loss}\;(\%)=\frac{\left(\mathrm{weight}\;\mathrm{of}\;\mathrm{raw}\;\mathrm{sample}\;-\;\mathrm{weight}\;\mathrm{of}\;\mathrm{cooked}\;\mathrm{sample}\right)}{\left(\mathrm{weigh}\;\mathrm{of}\;\mathrm{raw}\;\mathrm{sample}\right)}\times 100$$

### 2.9. Oxidative Stability Measurements of Rabbit Meat

The thiobarbituric acid-reactive substances (TBA-RS) assay has been widely used to measure lipid peroxidation in stored animal products. It is often considered a good indicator of their levels of lipid peroxidation, described by Djenane et al. [19]. The assay involves the reaction of lipid peroxidation products, primarily malondialdehyde (MDA), with thiobarbituric acid (TBA), which leads to the formation of MDA-TBA-RS. TBA-RS yields a red–pink color that can be measured spectrophotometrically at 532 nm. The TBA-RS value was expressed as mg MDA/kg of rabbit meat sample and calculated using a standard curve prepared with 1,1,3,3-tetramethoxypropane (Sigma Aldrich Corporation, St. Louis, MO, USA).

### 2.10. Determination of Total Volatile Basic Nitrogen in Rabbit Meat

The progression of rabbit meat protein degradation was evaluated through the total volatile basic nitrogen (TVBN) determination and was measured by semi-micro steam distillation. A sample of 10 g minced rabbit meat was dispersed in 100 mL distilled water and stirred for 30 min, and the mixture was then filtered. The TVBN value was determined according to the consumption of hydrochloric acid and calculated using the following equation:TVBN (mg/100 g) = (V_1_ − V_2_) × C × 2800(3)
where V_1_ is the titration volume of hydrochloric acid in the sample, V_2_ is the titration volume in the blank, and C is the concentration of hydrochloric acid (0.01 mol/L).

### 2.11. Microbiological Analysis of Rabbit Meat

On each sampling day, 25 g of rabbit meat samples were aseptically weighed into a sterile plastic stomacher filter bag (Seward) and diluted with 225 mL of buffered peptone water (CM1049). The content was homogenized in the stomacher (Inter Science, Saint Nom, France) for 2 min at room temperature. The resulting slurries were serially diluted in sterile peptone water. Sample dilutions were spread and plated on appropriate media in duplicate. The media used for enumeration of *C. jejuni* was *Campylobacter* blood-free selective medium (CCDA) (CM0739 plus selective supplement SR0155). The CCDA plates were incubated at 42 °C for 48 h in 2.5-L gas jars with CampyGen microaerophilic generating gas packs (CN0025). Counts of aerobic psychrotrophic bacteria were determined in plate count agar (PCA) (CM0463) and incubated at 7 °C for 10 days. *Brochothrix thermosphacta* was enumerated on streptomycin sulfate cycloheximide thallous acetate agar (STAA) (CM0881), supplemented with STAA Selective Supplement (SR0151) following 18 h of aerobic incubation at 26 °C. *Pseudomonas* spp. were enumerated in plates of Pseudomonas CFC selective agar (CM0559 plus selective supplement SR0103) which were incubated at 25 °C for 48–72 h. The logs of mean values for the counts from triplicate plates were recorded. Counts were expressed as the log_10_ of colony-forming units (CFU) per gram (log_10_ CFU/g).

### 2.12. Panel for Sensory Evaluation of Rabbit Meat

All assessments were carried out in respecting the safety measures on social distancing imposed by the COVID-19 emergency. Preparation of rabbit meat samples for sensory analysis was performed after the packages containing the carcasses were opened and held for 1 h at 4 °C under aerobic conditions. Panel tests were performed on rabbit meat samples every 5 days for a total of 20 days of refrigerated retail display in order to test their visual appearance, olfactory acceptability, and taste. Three samples from each group were taken at each selected time. The analysis was carried out by six trained panelists from laboratory staff (Laboratory of Meat Quality and Meat Safety, University of Tizi-Ouzou, Tizi-Ouzou, Algeria) who evaluated on a 1 to 5 scale the following parameters: meat odor intensity, spicy odor intensity, and color intensity. The rating for each sample ranged from none (scale of 1) to extremely (scale of 5) as described by Djenane et al. [20]. A score value >3 of any attribute (odor and spicy), denoted that rabbit meat was not acceptable by panelists. On the contrary, for overall acceptability, a score value < 3 denoted that rabbit meat was not acceptable to panelists.

### 2.13. Statistical Analysis

All experiments were performed in triplicate and the values were represented by the mean ± standard deviation (SD). The results were analyzed by ANOVA test using the Statistical Package for the Social Sciences software (SPSS version 21, IBM Corporation, Armonk, NY, USA). The effect of the marinated solutions treatment and low-O_2_ MAP on shelf life and microbiological safety of packaged rabbit meat was evaluated during display period and the level of significance was set at *p* < 0.05.

## 3. Results and Discussion

### 3.1. Changes in the pH Value of Rabbit Meat

Figure 2 shows that the pH in rabbit meat at day 0 was 6.12. The pH value of the group of unmarinated samples stored, respectively, under aerobic (1-A) and microaerophilic conditions at low-O_2_ (1-MA) increased gradually from day 0 to day 10 and then continued to increase slightly during the remainder of the retail display period. On the contrary, the pH value of the marinated groups (2-A and 2-MA) did not change significantly (*p* < 0.05) from day 5 to the final day of the retail display period.

Commonly, an increase in pH value indicates the degradation of rabbit meat quality during storage, probably due to the production of basic substances (amines, ammonia) through the decomposition of proteins [21], and by the presence of large quantities of biogenic amines, which were produced by the fast development of microorganisms [22,23]. Similar findings were also reported in poultry meat by Yücel et al. [24] and Yusop et al. [25]. Wang et al. [26] observed that Sichuan pepper *Zanthoxylum bungeanum* essential oil exhibited a strong inhibition effect on the increase in pH during 12 days of rabbit meat storage due to antimicrobial properties.

In the first hours after slaughter, the glycogen in the muscles is converted into lactic acid, which leads to acidification of the meat. Generally, the final pH value of meat can be affected by the pre-slaughter stress of the animal, which is negatively correlated with the sensory quality of the product. Accelerated pH decline and low ultimate pH are related to the development of low water-holding capacity and unacceptably high purge loss. Rapid pH decline while the muscle is still warm causes the denaturation of many proteins, including those involved in binding water. The most severe purge or drip loss is often found in PSE (Pale, Soft, and Exudative) meat when the pH of muscle drops below 5.8 during the first hours after slaughter while the rabbit carcass is still hot (~35 °C) [27].

### 3.2. Oxidative Stability of Rabbit Meat (TBA-RS)

Rabbit meat is very rich in PUFAs, and therefore is considered very vulnerable to oxidative reactions [28,29]. Lipid oxidation constitutes one of the main causes of reduced shelf life of foods and has been implicated in the formation of rancid odor, discoloration, abnormal flavors [30], and worse, produces toxic compounds such as MDA, cholesterol oxidation products, and free radicals that can affect the health of consumers [31].

The secondary oxidation product such as MDA is widely evaluated by TBA-RS assay. The TBA-RS value is often used as an efficient biomarker of lipid oxidation in animal products [16,32,33,34]. Figure 3 shows that the oxidative stability of the rabbit meat was affected (*p* < 0.05) by marinade treatments and retail display conditions. It is shown that TBARS values significantly increased (*p* < 0.05) during the first 10 days of display for the samples packed under aerobic conditions (1-A and 2-A), and this increase coincides with the decrease in sensory quality of the displayed product due to excessive development of spoilage microorganisms and rancid off-odor.

At the end of the display (20 days), the samples stored under aerobic conditions 2-A showed TBA-RS values (2.75 mg MDA/kg) higher than the groups 1-MA and 2-MA which were stored under a microaerophilic atmosphere (1.72 and 1.12 mg MDA/kg, respectively) (*p* < 0.05). The lack of data (not determined) at an early stage (15 and 20 days) for 1-A samples is due to excessive development of spoilage microorganisms and rancid off-odor.

The MDA value of rabbit meat on days 0 and 5 was not significantly different (*p* > 0.05) between samples and remained below 1 mg MDA/kg, but from day 5 onwards the values increased greatly in carcasses stored under aerobic conditions, while they remained lower in samples packed in MAP. Likely, the intrinsic antioxidant *post mortem* defense system of muscle remains active for a few days *post mortem* [35,36], clarifying why statistical differences in TBARS values were only evident for all samples after 5 days of display. Wang et al. [28] and Nakyinsige et al. [37] also found that the contents of TBA-RS steadily increased during retail display of refrigerated rabbit meat. In agreement with the present data, Djenane et al. [16,38], Aboudaou et al. [39], and Ait Ouahioune et al. [4,5] reported high oxidative stability in the stored animal products treated with natural bioactive products. Wenjiao et al. [40] mentioned that TBARS values of 2 mg MDA/kg in meat could be considered as the rancidity detection limit by the consumer. Likewise, in work carried out on the effect of active packaging under high O_2_ on the freshness of beef meat, significant correlations were identified between TBA-RS, microbial loads, and sensory panel assessment [3]. Based on these correlations, the authors identified TBA-RS levels of 2 mg MDA/kg as the point of sensory panel rejection and suggested this as a critical limit. In the present study, the TBARS values in marinated (2-MA) or unmarinated (1-MA) rabbit meats stored in microaerophilic atmospheres were 1.12 and 1.72 mg MDA/kg, respectively, and did not exceed the threshold levels (2 mg MDA/kg) even at the end of the display period (20 days). Therefore, the oxidative evolution during the display period becomes a valuable indicator of the antioxidant effect of these treatments. This parameter reveals that the marination of samples combined with packaging under the microatmosphere is particularly effective in maintaining oxidative stability (*p* < 0.05). Other recent studies of fresh meat and marinated meats (multiple ingredient inclusion) delayed meat lipid oxidation for up to several days during its retail display [23,24,41].

### 3.3. Microbiological Analysis of Rabbit Meat

Fresh meat is subject to microbial spoilage during storage, resulting in discoloration, off-flavors and off-odors, loss of nutrients, and changes in texture, as well as potential health hazards. Microbiological analyses have been aimed at identifying several microbiological groups commonly associated with the spoilage of stored meats. The development of bacterial growth in rabbit meat samples during exposure is presented in Table 1.

Our results show an adequate initial microbiological quality of the rabbit carcass used in the present work, indicating that the appropriate hygienic conditions for processing the carcasses were followed. In fact, the bacterial load of the main spoilage bacteria was less than 3.6 log_10_ CFU/g. In general, the main spoilage bacteria associated with chilled meat are *B. thermosphacta*, lactic acid bacteria (LAB) under low-O_2_ conditions, and *Pseudomonas* spp. during storage under aerobic conditions. Rodriguez-Calleja et al. [42] found that the spoilage microbiota of rabbit meat in 100% CO_2_ and vacuum was dominated (up to 54%) by lactic acid bacteria (LAB). Under a gas mixture atmosphere of 35% CO_2_/35% O_2_/30% N_2_, *Pseudomonas*, LAB, and *Brochothrix thermosphacta* accounted for 8.7%, 4.0%, and 2.4%, respectively.

The initial counts of aerobic psychrotrophic bacteria, *Pseudomonas* spp., and *B. thermosphacta* in the rabbit samples were 3.5, 2.02, and 2.35 log_10_ CFU/g, respectively (Table 1). However, as exposure continued, all bacterial counts showed significant increases (*p* < 0.05). Similarly, Wang et al. [28], Nakyinsige et al. [36], and Djenane et al. [43,44,45] found that the count of spoilage bacteria increased steadily during the storage of rabbit and other meat. Total psychrotrophic bacteria (TPB) counts during the first 5 days in non-marinated samples stored under aerobic conditions (1-A) were above (*p* < 0.05) in all other groups, reaching 6.4 log_10_ CFU/g, and were the only ones that exceeded 7.0 log_10_ CFU/g at 10 days of exposure (7.6 log_10_ CFU/g). However, TPB was significantly (*p* < 0.05) lower in samples packed with low-O_2_ (1-MA and 2-MA) compared to aerobic samples. The largest differences were observed from day 10 to the end of exposure, when TPBs were 3.12 and 4.64 log_10_ CFU/g vs. 7.59 and 6.39 log_10_ CFU/g, respectively. At the end of exposure (20 days), TPB, *Pseudomonas*, and *B. thermosphacta* count in the marinated samples packed in the low-O_2_ atmosphere (2-MA) reached 4.89, 2.09 and 3.72 log_10_ CFU/g, respectively. At the same time, in the non-marinated samples (1-A) values of “nd—due to excessive development of spoilage microorganisms and a noticeable unpleasant rancid off-odor—, 7.89 and 4.58 log_10_ CFU/g, respectively, were obtained, and the differences were significant (*p* < 0.05).

Therefore, the marinating process with *Ras El-Hanout* combined with a low-O_2_ atmosphere was very effective and guaranteed the microbial stability of the rabbit meat during its display. The presence of a low-O_2_ level would be sufficient for the growth of aerobic meat spoilage bacteria, although their growth can be retarded by incorporating 40% CO_2_ in the gas mixtures. In rabbit meat, Rodriguez-Calleja et al. [42] evaluated the effect of different modified atmospheres on storage stability. The authors observed that the best atmosphere for preserving the microbiological quality of rabbit meat and extending its shelf life was 100% CO_2_ (35 days), followed by vacuum packaging (VP) (~30 days) and 35% CO_2_/35% O_2_/30% N_2_ mixture (~20 days).

The ICMSF (International Commission for Microbial Specification in Food) recommends ~10^7^ CFU/g for meat as an acceptable threshold [46]. In the unmarinated samples packed in aerobic conditions (1-A), and according to the microbiological guidelines and specifications, TPB exceed the limit (~7 log) after the 5th day of display, indicating that the end of microbiological shelf life was reached after only 5 days for this group. However, the shelf life of marinated samples packed in a low-O_2_ atmosphere (2-MA) was extended even until the end of the display (20 days). However, this recommended value was also never reached in unmarinated samples packaged under low-O_2_ (2-MA), even during the 20 days of display. The observed results could be attributed to the combined effects of different bioactive molecules present in the marinade solution, among others the polyphenols (535.29 mg GAE/g), which are already recognized for their antimicrobial activity, combined with the presence of 40% CO_2_ in the gas mixture. Lipophilic polyphenol compounds can act at the bacterial membrane level disrupting its functions and then breaking and leaking the cytoplasmic content.

A similar trend was observed by Siroli et al. [41], who indicated that marinated pork meat presented a lower altering microbial load compared to unmarinated samples, from the sixth to the thirteenth day of storage. Several studies have highlighted the broad spectrum of antimicrobial effect of Mediterranean species against a high number of microorganisms [47,48,49], which has been established and extended in the present work. The same trend was observed by Osaili et al. [10], who studied the antimicrobial effect of plant extracts (EOs or specific compounds) on spoilage microorganisms in marinated and vacuum-packed camel meat during storage at 4 and 10 °C. Similarly, Haute et al. [50] studied the effect of thyme EOs and specific compounds on the shelf life of marinated fish and meat, concluding that this method constitutes a potential tool to curb spoilage microorganisms. Similarly, Mancini et al. [51] tested garlic on rabbit burgers and found antibacterial effects against total aerobic psychrotrophic bacteria and *Pseudomonas* spp. during one week of storage.

However, the degree of antimicrobial activity depends on the ingredients of the marinade mix, the level of initial microbial load, the pH of the matrix, and other storage conditions (temperature and type of packaging). Furthermore, the effect of the herbal extracts was often only achieved with large amounts, which may imply an organoleptic influence on the food matrix. Strategies to supplement these natural bioactive products in rabbit feed have been tested. Kone et al. [7] evaluated the effect of dietary supplementation combined with extracts of onion (500–1000 ppm), cranberry (500 ppm), and olive oil (100 ppm) on the quality of rabbit meat during storage. This supplementation allowed better microbial stability against *Pseudomonas* and the total aerobic mesophilic bacteria. Some *Pseudomonas* species capable of growing in food at refrigeration temperatures (0–7 °C) produce colored pigments on the surface of rabbit carcasses stored aerobically at refrigeration temperatures [28,52,53].

### 3.4. Color Stability of Rabbit Meat

For consumers, color has been described as one of the main characteristics of fresh meat at the point of purchase, and color certainly plays a key role in the presentation, appearance, and acceptability of rabbit meat. Yusop et al. [25] found that consumers considered surface color as contributing to the acceptability of marinated chicken. At the level of the refrigerated display showcases of supermarkets, maintaining the fresh color of rabbit meat during retail display is helpful for increasing sales because discoloration is a substantial economic matter for the meat industry due to shelf life limits [16]. Table 2 shows the effect of the marination, microaerophilic atmosphere packaging, and retail display days on the rabbit meat color (L*, a*, and b*).

The initial mean values of L*, a*, and b* of non-marinated fresh rabbits (1-A and 1-MA) were 62.85, 3.48, and 6.28, respectively. The red color is the most widely preferred color and mostly indicates the freshness and healthiness of retailed meats, whereas brown indicates a deficiency for both attributes. Regarding, the lightness (L*) of meat, its acceptability is dependent on ethnic aspects and consumption behaviors. The use of *Ras El-Hanout* marination exerted a significant influence on the redness (a*), yellowness (b*), and lightness (L*). The higher a* and b* values obtained for marinated rabbit samples during the whole period of display might be probably due to the existence of coloring compounds in the marinade solution itself (i.e., cumin, sweet red pepper, coriander, curcuma, EVOO, and concentrated lemon juice). These compounds might have increased the redness and yellowness of rabbit meat samples, and this constitutes a great advantage from a marketing point of view. A similar phenomenon was reported for marinated pork loin slices vacuum-packaged by the addition of oil/lemon/beer with the inclusion of juniper, oregano, and rosemary essential oil [41]. In terms of color, the scientific literature confirms that fresh rabbit meat tends to lose considerable brightness and redness during storage, and these losses occurred in parallel with an increase in yellowness [28]. A similar finding has been reported in beef during retail display [17]. Moreover, in agreement with earlier studies, a fall in L* and a* was often observed with a concomitant increase in TBARS formation and microbial development, which suggests that the oxidation of PUFAs contributes to the enhanced oxidation of myoglobin (Mb). Wang et al. [54] studied in a model system containing compounds native to rabbit meat the effect of MDA on the color stability of this system. These authors observed an increase in the concentration of metmyoglobin (MetMb) due to the presence of MDA. In addition, MDA acted as a precursor in the formation of reactive oxygen species (ROS) and in the release of free iron (Fe) (non-heme iron) from the heme system.

Our previous studies have already shown an underlying relationship between meats’ lipid oxidation and color change during storage [38]. As well, lipid oxidation phenomenon, protein oxidation, and microbial spoilage are other major causes underlying the color deterioration of meat. Wang et al. [28] suggested that these three factors conjointly affected rabbit Mb autoxidation and consequently the meat color stability during retail display. Fresh rabbit meat yellowness (b*) has been shown to be closely related to lipid oxidation [21,55] and, accordingly, a significantly positive association between the b* value and TBARS. Generally, microbial development is concomitant with O_2_ consumption, resulting in reduced partial O_2_ pressure (PaO_2_) on the meat’s surface. For its part, this reduction can be at the origin of the oxidation of the pigment Mb [28,56]. Recently, Redondo-Solano et al. [57] found that the color stability of stored ground rabbit meat was improved using vacuum storage compared to simple overwrapping. In light of our results, we can conclude that rabbit meat discoloration during exposure was significantly (*p* < 0.05) influenced by microbial spoilage, lipid oxidation, and possibly to a large extent by Mb oxidation. Several authors have pointed out the role of temperature in the preservation of rabbit meat. Retailing at a temperature of −1 °C, compared to +1 °C and +8 °C, favored the color stability of rabbit meat by delaying lipid oxidation, Mb oxidation, and microbial growth [58]. This delay in the deterioration of color would positively influence the purchase intention of consumers, so it can be promoted as a retail sale temperature for the commercialization of rabbit meat. It is also very important to point out that to promote the color stability of marinated rabbit meat, strategies inhibiting the formation of TBARS, TVBN, and the loss of color need to be explored. In this context, the effects of promising technologies and alternative rabbit diets with antimicrobial and antioxidant properties have already been determined for their ability to preserve the color of retailed fresh rabbit meat [59,60,61].

### 3.5. Total Volatile Basic Nitrogen of Rabbit Meat

Total volatile basic nitrogen (TVBN) mainly results from the degradation of proteins by the action of bacteria or enzymes present in meat and is widely used to assess the deterioration of animal products during storage. The chemical stability of the rabbit meat was significantly affected (*p* < 0.05) by marinade treatments and retail display conditions. As depicted in Figure 4, TVBN values significantly increased (*p* < 0.05) during the first 10 days of the display, especially for the unmarinated samples packed under aerobic conditions (1-A). The TVBN values in this group increased from 6.46 to 27.1 mg/100 g (4.2-fold), coinciding with the decrease in sensory quality of the stored product within the same display period due to excessive development of spoilage microorganisms (PCA count) and rancidity (TBARS values). However, marinated samples under aerobic conditions (2-A) showed a moderate increase to 15 mg/100 g during the same period.

After 10 days of storage, the TVBN value of group 1-A was higher than 30 mg/100 g indicating a frankly unpleasant off-odor. In the samples of group 2-A, the threshold value of 30 mg/100 g was reached at 20 days. However, samples packaged in low-O_2_ modified atmospheres (groups 1-MA and 2-MA) maintained TVBN values around 15 mg/100 g throughout the retail display period. This parameter reveals that the marination of samples combined with packaging under the microatmosphere is particularly effective in preventing the formation of degradative nitrogenous compounds (*p* < 0.05).

TVBN can be used as a biomarker to assess the microbial spoilage of animal products, due to its connection with the activity of spoilage bacteria, which degrade nitrogenous compounds. For example, Chen et al. [62] found a positive correlation between microbial growth and TVBN values in pork meat during storage and indicated that values of TVBN around 15 mg/100 g are indicative of the freshness of the stored product. The results obtained here suggest that the marination process with incorporation of *Ras El-Hanout* combined with low-O_2_ packaging was the most antimicrobial treatment in rabbit meat against the main spoilage microorganisms as compared to control samples. Wang et al. [28] and Redondo-Solano et al. [57] found that the TVBN content steadily increased during the retail display of rabbit meat. The Egyptian Organization for Standardization and Quality Control (EOS 1090/2005) has set guidelines and thresholds to define the freshness or spoilage of rabbit meat. According to this organization, rabbit meat is considered fresh when the TVBN concentration is less than 20 mg/100 g. In accordance with our study, Mansur et al. [63] found that the TVBN values exceeded freshness guidelines (>20 mg/100 g) in beef meat when stored under aerobic conditions compared to vacuum-packaged beef for 9 days of storage. TVBN values of 25–28 mg/100 g were also proposed by Senapati and Sahu [64] as an acceptable threshold for chicken meat. Khulal et al. [65] considered TVBN values above 15 mg/100 g in chicken meat to be indicative of loss of freshness, while Ghollasi-Mood et al. [66] suggested TVBN values close to 25 mg/100 in chicken meat stored under refrigeration as unacceptable. According to Bekhit et al. [67], the degree of spoilage in different tested meats is doubtful to be conceived by the assignment of a sole TVBN acceptable threshold. The relationship between TVBN and microbial growth is evident in studies aimed at prolonging the shelf life of meats by inhibiting microbial growth, whose acceptance limit value would be around 7 log_10_ CFU/g [58].

The increase in pH during storage has been often found to have a strong positive correlation with the amount of TVBN [64,68]. As previously indicated, the pH value of the marinated groups (2-A and 2-MA) did not change significantly (*p* < 0.05) during the retail display period despite the addition of concentrated lemon juice, probably due to *post mortem* muscle buffering capacity [69]. In addition to all that, rabbit *antemortem* practices that influence meat pH will play an important role in determining the final pH and the concentration of carbohydrates accessible for bacteria prior to being forced to metabolize protein molecules for energy. Thus, Sun et al. [70] found a positive correlation between pH and TVBN linked to a good environment for the proliferation of spoilage bacteria at higher pH values. High pH facilitates the gradual transition from carbohydrate (glucose, glycogen) dependent bacteria to protein-degrading bacteria. Umuhumuza and Sun [71] observed that a fast increase in the TVBN amount from 6 to 13.5 mg/100 g between the 2nd and 3rd day of storage at 4 °C was found upon the depletion of *post mortem* glucose in meat. Adding preservative compounds such as kojic acid, ε-poly-l-lysine, sodium diacetate, or potassium sorbate for stabilizing pH of pork meat stored under vacuum packaging allowed TVBN level below 12.5 mg/100 g after 21 to 42 days of storage [72].

### 3.6. Warner–Bratzler Shear Force of Rabbit Meat

The main attribute that consumers associate with the palatability of good quality meat is tenderness. The effect of marinade solution and low-O_2_ atmosphere packaging on the instrumental texture of rabbit meat is shown in Figure 5. Rabbit meat from all samples was found to be initially tender with relatively lower shear force values (3.06 kg). On the 5th day of display, a decrease in shear force was observed for marinated samples (2-A and 2-MA) compared to unmarinated ones (1-A and 2-A), and this decreasing trend was maintained throughout the exposure period. In all samplings, marinated meats were significantly more tender than unmarinated meats (*p* < 0.05). However, no differences were observed in shear force due to the type of packaging, either in air or in low-O_2_ modified atmospheres.

These results suggest the usefulness of an acidic marinade solution to improve the tenderness of rabbit meat, as previously reported by Pérez et al. [73], Gomez-Salazar et al. [74], and Simitzis et al. [75], probably due to enhanced proteolysis [9,76]. Ertbjerg et al. [77] found that injection of a lactic acid solution into beef muscle in its pre-rigor state could tenderize it. Injection of lactic acid significantly reduced shear force by increasing the activity of lysosomal cathepsins. Regarding the effect of culinary treatments on the tenderness of rabbit meat, Dal Bosco et al. [78] found that compared to fresh meat (3.59 kg) shear force increased with roasting and frying, while it decreased in boiled meat.

Several studies have been undertaken in order to find threshold values of meat instrumental texture (shear force) for sensory tenderness acceptability. Correlations of instrumental texture (WBSF) with the sensory panel assessment are known to be highly variable. Destefanis et al. [79] reported a correlation of −0.72. Similarly, Caine et al. [80] obtained correlations ranging from −0.32 to −0.94. In the beef industry, packaging with high O_2_ concentration has been used for many years [81,82,83]. However, this packaging system can negatively affect both shear force and sensory tenderness, perhaps due to the phenomenon of myofibrillar protein cross-linking and protein oxidation [84]. Similarly, Grobbel et al. [85] observed that beef stored under a vacuum or in the presence of 0.4% CO is more tender compared to beef packaged in a modified atmosphere with high O_2_ concentration. However, in our case, the use of a low-O_2_ atmosphere (microaerophilic atmosphere) saved rabbit meat from this negative phenomenon of tenderness.

The initial shear force values obtained by other researchers are slightly higher than those obtained in the present study (3.06 kg). These differences may be due to the breed of rabbits, housing conditions, sex, type of feed, slaughter weight, age at slaughter, and meat condition (fresh meat, raw meat, cooked meat…) [86,87]. By comparison, Vitale et al. [88] observed higher values of shear force (5.10 kg) for beef a few hours *post mortem*. Destefanis et al. [79] reported that most panelists perceived the meat as tough when WBSF values were >5.37 kg and tender when WBSF values were <4.37 kg, respectively. Huffman et al. [89] already proposed WBSF values of 4.10 kg as a threshold of satisfactory eating quality for consumers. In the experimental group that did not receive marinade treatment, its shear force was >2 kg during the entire exposure period and was statistically higher than that of the marinated meat (*p* < 0.05), which had a shear force < 1 kg after 10 days and until the end of retail display.

### 3.7. Water Holding Capacity and Cooking Loss of Rabbit Meat

Meat appearance, particularly loss of exudates and color degradation, determines how consumers recognize the quality of the exposed product and affect purchasing behavior. Table 3 presents variations in water holding capacity (WHC) expressed as water released (RW%) and cooking loss (CL%) over the exposure of rabbit carcasses. The results indicated that the marination process using the *Ras El-Hanout* blend of spices improved the water-holding capacity of meat. Thus, after 5 days of exposure, marinated samples showed better WHC as compared to non-marinated samples. Additionally, the exposure period had a significant effect (*p* < 0.05) on the WHC of unmarinated samples, but not on that of marinated samples during 20 days of exposure. Differences between groups were observed from day 10 to the end of exposure, with significantly lower RW% and CL% values (*p* < 0.05) in the marinated samples than in the unmarinated ones.

In the scientific literature, the WHC of meat varies considerably. Different authors use different methodologies and ways of expressing the results. The use of different pressures and different filter papers often results in different amounts of water extruded from meat. In addition, the differences are influenced by the weight of the rabbits at slaughter. A decrease in CL percentage can cause meats to become unacceptable due to loss of nutrients during exposure, which is of great importance to consumers. The results regarding CL at different exposure periods are shown in Table 3. On the 5th day of exposure, a decrease in CL was observed in marinated samples (2-A and 2-MA) compared to non-marinated samples (1-A and 1-MA). At 10 days of exposure, the CL of the marinated samples decreased by 44.54% (14.17 vs. 25.55) and 28.54% (15.25 vs. 21.34), respectively. Similarly, and throughout the remaining display period (15 and 20 days), marinated samples remained significantly moister (*p* < 0.05) than non-marinated samples packaged under aerobic conditions (1-A). These results suggest the usefulness of an acidic marinade solution to improve the moisture of exposed rabbit samples, as previously reported by Gao et al. [90] and Siroli et al. [41] in vacuum-packed marinated pork. This trend could possibly be due to the increased absorption and gradual retention of the marinade solution throughout the exposure period, which could have resulted in less fluid loss during cooking. It is also likely that this improvement is due to the beneficial effect of the marinade solution, which contains bioactive antioxidant molecules, on the integrity of the muscle fibers, thus increasing their ability to retain water due to membrane stability. Contrary to our observations, Siroli et al. [41] found that after 2 weeks of storage, marinated pork samples presented a slightly higher cooking loss compared to control samples. Among the different cooking methods for rabbit meat, grilling seems to be the most favorable method to avoid cooking losses (31.5 vs. 37%) [91].

With marination, a more fragrant meat is obtained after cooking, with a more pleasant texture, more tender, and juicier. In fact, in the case of cooked rabbit meat, the marinade will prevent the product from drying out. Moreover, compared to samples stored in a modified atmosphere with low-O_2_ content, aerobically stored samples showed a lower WHC during the whole exposure period. This phenomenon may have been the result of more severe proteolysis under aerobic conditions caused by the activity of bacterial growth and the different oxidation reactions that may have taken place during the exposure of the product. In our study, the use of a microaerophilic atmosphere does not appear to have an adverse effect on exudate loss. However, it has been observed that reduced acceptability of rabbit meat was associated with exudate losses when packaged in an atmosphere with 100% CO_2_ or higher O_2_ [42,82,83].

### 3.8. Sensory Attributes Evaluation

One strategy of the food industry to diversify products and increase the attractiveness of meat products for consumers is the addition of herbal extracts and spices. However, their sensory compatibility and their effect on the sensory profile of the final products must be taken into account. In the present work, we previously carried out an optimization of all the ingredients added to the marinade solution because the original compounds of the *Ras El-Hanout* blend have their own characteristic aroma, which could mask the perception of the unpleasant odor that may develop in the exposed product with the passage of time. Table 4 shows the assessment of spicy intensity, odor intensity, and overall acceptability on a 5-point scale with the range of none (scale of 1) to extremely (scale of 5) as means ± standard deviation. A score ≥ 3 in spicy and odor intensity attributes denoted that rabbit meat was unacceptable. For overall acceptability, a score < 3 denoted that the rabbit meat was unacceptable to the sensory panel.

According to our study, the data show significant differences (*p* < 0.05) between the tested treatments. Marinated rabbit meat presented better spicy and odor intensity scores than unmarinated rabbit meat. In addition, the marinated rabbit samples showed better overall acceptability, clearly perceived by the sensory panelists during the entire display period. It is noteworthy that the increase in a* (redness) and b* (yellowness) values (Table 2)—due to the existence of coloring compounds in the marinade solution itself—did not negatively affect the sensory evaluation of the marinated samples by the panelists. In addition, the texture parameters (shear force) of the rabbit carcasses were positively affected by the marinade treatment, resulting in an overall improvement in suitability compared to the unmarinated samples. These results are in agreement with those of previous studies, according to which marinating treatments improve the tenderness and juiciness of beef and poultry meat, respectively [92,93].

In another study, Castrica et al. [94] found that the panel assigned a higher score to meat from rabbits fed goji berry as a dietary supplement and purchase awareness increased when the rabbit diet was previously identified. Similar results had already been highlighted by Perna et al. [95], who suggested that a diet enriched with cauliflower (*Brassica oleraceae* var. *botrytis*) leaf powder was a good strategy to produce higher-quality rabbit meat. In addition, the sensory quality of rabbit meat could be associated with genetic type, breeding techniques, age at slaughter, and muscle type. Siroli et al. [41] evaluated the combined effects of marinade solution (beer/lemon/oil) with essential oils on the quality and safety of packaged pork. The authors suggested that marinating improved the tenderness of meat samples, perceived more positively by the panelists. In contrast, Meineri et al. [96] found no improvement in the sensory attributes of rabbit meat fed with chia (*Salvia hispanica*). A recent study by Rossi et al. [97] and Al Jumayi et al. [6] noted an improvement in the sensory attributes of meat from growing rabbits fed a mixture of plant polyphenols.

### 3.9. Shelf Life

To extend the shelf life of rabbit meat during retail exposure, correlations between meat sensory properties, oxidation reactions (lipids, pigments, and proteins), microbial spoilage, and safety need to be thoroughly controlled. Display of rabbit carcasses showed that treatment with marinade solution and low-oxygen atmosphere packaging increased product shelf life (*p* < 0.05) (Table 5). Shelf life was assessed by detection thresholds (limits) for microbial spoilage (total psychrotrophs > 7 log_10_ CFU/g), oxidative reactions (TBA-RS > 2 mg MDA/kg), TVBN (>20 mg/100 g) and overall acceptability (<3 points), during the entire display period. Samples packaged under low-O_2_ atmosphere (1-MA and 2-MA) did not deteriorate prominently, as indicated by the high retail shelf life, which was extended by more than fifteen (>15) additional days compared to control samples (1-A). As we have indicated in the preceding sections, many variables (overall acceptability scores, microbial counts, CIE Lab values, TVBN, TBARS) differed between exposure times, packaging systems, and marinade treatments (*p* < 0.05). In particular, meat discoloration reflects a deficiency of freshness and wholesomeness and is associated with shelf life, making it an important economic issue for the meat industry. The industry must preserve as much of the fresh color of rabbit meat as possible during retail displays to increase sales.

Nakyinsige et al. [36] found that spoilage of stored rabbit meat is primarily due to protein and lipid oxidation as well as microbial spoilage. Protein oxidation induced by reactive oxidative species (ROS) leads to protein structural changes [98], with undesirable effects on meat tenderness, WHC, and color [99]. Protein and lipid oxidation coexist in the meat system and mutually enhance each other. Malondialdehyde (MDA) is the key by-product of PUFA oxidation, and some researchers have observed that lipid oxidation products could induce protein oxidation [28,100]. In addition, TVBN concentration is commonly used as an indicator to assess microbial spoilage of rabbit meat because of its connotation with the shelf life of the product during storage.

This study highlighted that treatment with a marinated solution containing *Ras El-Hanout* is an effective approach to producing rabbit meat with improved technological and functional quality. Some studies have described that natural antioxidants may be associated with meat color and lipid stability, improved antioxidant status, and shelf life, through various mechanisms such as metal chelators, radical scavengers, and quenchers of free singlet oxygen (^1^O_2_) [19]. Śmiecińska et al. [101] studied the shelf life of rabbit meat burgers treated with *Allium* species powders (garlic and ramsons) and estimated significant shelf life extension and improved sensory properties of oven-baked burgers. The packaging with low-O_2_ atmospheres also contributed to the extension of shelf life. For comparison, Pereira and Malfeito-Ferreira [102] studied the shelf life of rabbit carcasses stored under aerobic refrigerated (at 4 °C) conditions and reported a shelf life of 7 days.

The interrelationship between lipid, protein, and Mb oxidations in refrigerated stored rabbit meat has been reported in several papers. Wang et al. [27] observed that *Zanthoxylum bungeanum* essential oil (ZBEO) showed a strong inhibitory effect on microbial growth, protein and lipid oxidation, TVBN formation, and discoloration of rabbit meat during 12 days of storage, and concluded that ZBEO is a potential natural preservative to promote the shelf life of rabbit meat during storage. Meneses and Teixeira [15] published a review article on the margination of poultry meat products and found that marinating processes, in addition to reducing pathogenic microorganisms, have a significant ability to extend the shelf life of products. It is important to note that several plants may contain substances that can be of health concern when used in food. The European Food Safety Authority has warned of potential safety concerns of some botanicals that may require further investigation [103]. Therefore, the synergism between marinade treatment and other hurdles to spoilage microorganisms should be taken into account. The role of temperature, which can have a significant corrective effect on product shelf life, should not be underestimated. For example, Wang et al. [28] found that a storage temperature of −1 °C may be suitable as a commercial retail temperature for rabbit meat because of its role in prolonging product stability and shelf life.

### 3.10. Challenge Trial with Campylobacter jejuni

Numerous studies have revealed a high prevalence of foodborne pathogens in meats due to poor hygiene during various operations. In recent years, surveillance programs in some developed countries have revealed a high level of *Campylobacter* contamination in poultry and poultry products. *Campylobacter* is the most common cause of foodborne bacterial zoonoses, with a steady increase in the number of cases. According to the European Food Safety Authority, up to 80% of human campylobacteriosis is attributed to “poultry” as a reservoir [104]. In Algeria, it is common for chickens and rabbits to be raised together with the consequent risk of *Campylobacter* transmission between the two species. On the other hand, the consumption of undercooked rabbit meat is recognized as a major risk factor for consumers at the level of roadside restaurants in the coastal area of Cap Djinet (Algeria). In this Mediterranean region known for its high tourist influx, restaurants often serve grilled whole carcasses of rabbits from a mixed farmyard (rabbits/poultry). Additionally, *post mortem* handling of rabbit carcasses in restaurants and retail establishments is a particularly critical process because of the potential for cross-contamination with chicken carcasses and contact surfaces. This can lead to the spread and growth of pathogens in the product. Therefore, minimizing the presence of pathogens in the rabbit carcass is key to increasing consumer safety.

In order to evaluate the effects of the *Ras El-Hanout* marinating process and the type of packaging on the safety of rabbit carcasses, a challenge trial was conducted by inoculating the pathogen *C. jejuni* (2 × 10^5^ CFU/g) into whole rabbit carcasses. The initial *C. jejuni* count on day 0 was 5.3 log_10_ CFU/g. As shown in Figure 6, the recovery of *C. jejuni* gradually increased in unmarinated control carcasses throughout the exposure period, reaching ~7 log_10_ CFU/g at day 20. In contrast, marinated rabbit carcasses showed lower *C. jejuni* counts, 4.79 log_10_ CFU in those kept under aerobic conditions and 3.66 log_10_ CFU/g in those packed in low-O_2_ atmosphere, respectively. Therefore, *C. jejuni* counts in rabbit carcasses marinated and packaged in a low-O_2_ atmosphere were significantly lower (*p* < 0.05) than in the other treatments.

As already mentioned, the final marinade solution contained large amounts of polyphenols (535.29 mg GAE/g), which are already recognized for their antimicrobial activity. These results were consistent with those of our previous works in which olive leaf extract from different varieties of Algerian olive trees showed antimicrobial effects in turkey meat, raw *Halal* minced beef, and camel meat during display [16,45,105]. Olive leaf extract contains great quantities of polyphenols (216.5 mg GAE/g). Djenane et al. [43] evaluated the antimicrobial activity of Eos from *Inula graveolens*, *Laurus nobilis*, *Pistacia lentiscus*, and *Satureja montana* against *C. jejuni* inoculated in chicken meat. The tested EOs inhibited the growth of *C. jejuni* at 2× minimum inhibitory concentration (MIC) values and reduced the levels in retail chicken meats packaged in microaerobic atmospheres.

Several studies have been conducted to evaluate the antimicrobial activity of different marinade solutions on poultry meat. Karyotis et al. [106] conducted a study on chicken breasts marinated and vacuum-sealed at 4 °C for 18 h and then thermally processed, observing that marinating increased the thermal sensitivity of *Salmonella* spp. and *L. monocytogenes*. Birk et al. [107] studied the antimicrobial effect of organic acids and marination ingredients on the survival of *C. jejuni* in chicken meat and reported a noticeable antimicrobial effect in the product. Zakariene et al. [108] conducted a study on broiler wings containing thyme-based marinade and reported good activity against *C. jejuni* reducing 1.04 log_10_ CFU/g after 7 days at 4 °C. Moon et al. [109] found that *S. Typhimurium* was fully inhibited in chicken breast after immersion in teriyaki sauce with carvacrol and thymol for 7 days at 4 °C. Evrendilek et al. [11] reported that high hydrostatic processing (HHP) treatment on marinated chicken reduced *Salmonella* counts to undetectable level.

Some information is available regarding the great antimicrobial spectrum of the individual bioactive compounds and their synergisms. Kiprotich et al. [110] suggested that marinades with thyme oil exhibited greater antimicrobial activity against *Salmonella* than marinades only containing yucca extract and lemon juice. Sengun et al. [111] investigated the effect of the marinade with grape products (juice and pomace) in poultry meat against several foodborne pathogens (*Escherichia coli* O157:H7, *Listeria monocytogenes*, and *Salmonella* Typhimurium) and found that marination resulted in decrease below the detection limit of studied microorganisms. Temperature abuse is a common occurrence during handling meats [44] and, an extra degree of safety may be assured in marinated products. In this context, Juneja et al. [112] studied the effect of grapefruit extract and temperature abuse on the growth of *Clostridium perfringens* from spore inocula in marinated chicken packaged under vacuum conditions. The authors concluded that *C. perfringens* spores had lower growth rates after germination. Moreover, the chicken samples added with marinade did not undergo major changes in their qualitative characteristics.

## 4. Conclusions

The results of the present study underline that marinating whole rabbit carcass using a marinade solution combined with *Ras El-Hanout* spice mixture allows for achieving an overall improvement in the physicochemical, microbiological, and sensory quality of rabbit meat packaged in a modified atmosphere with low-O_2_ during its exposure at 7 ± 1 °C for 20 days. All variables studied revealed that, regardless of the retail display period, the shelf life of rabbit meat was determined by the combined effects of the growth of spoilage microorganisms, the formation of oxidation and degradative products, and sensory deterioration. Our results revealed that marinated rabbit carcasses showed fifteen additional days of shelf life as compared to control unmarinated samples. Additionally, marination allowed us to prevent the growth of *C. jejuni* inoculated in rabbit carcasses, thus increasing the microbiological safety of the rabbit meat.

In this study, the development of new rabbit-based products showed potential for commercial insertion, offering versatility to traditional rabbit meat by adding aroma, flavor and extending shelf life, and increasing the opportunity for use in several gastronomic preparations. With the marinade, we will obtain after cooking rabbit meat that is not only more fragrant but also has a more pleasant texture and is more tender and juicier. In fact, for cooked rabbit meats, the marinade will prevent the product from drying out. Considering these factors, marination and low-O_2_ packaging may be a valuable and relatively low-cost method for the stability and safety of rabbit carcasses in retail establishments.

## Figures and Tables

**Figure 1 foods-12-02931-f001:**
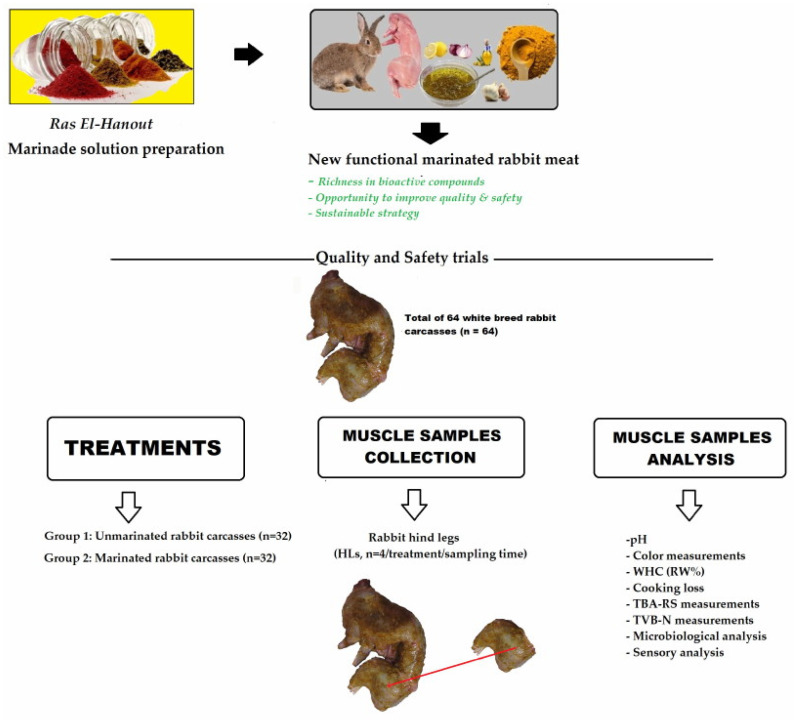
Sampling plan and muscle samples collection for the different analyses.

**Figure 2 foods-12-02931-f002:**
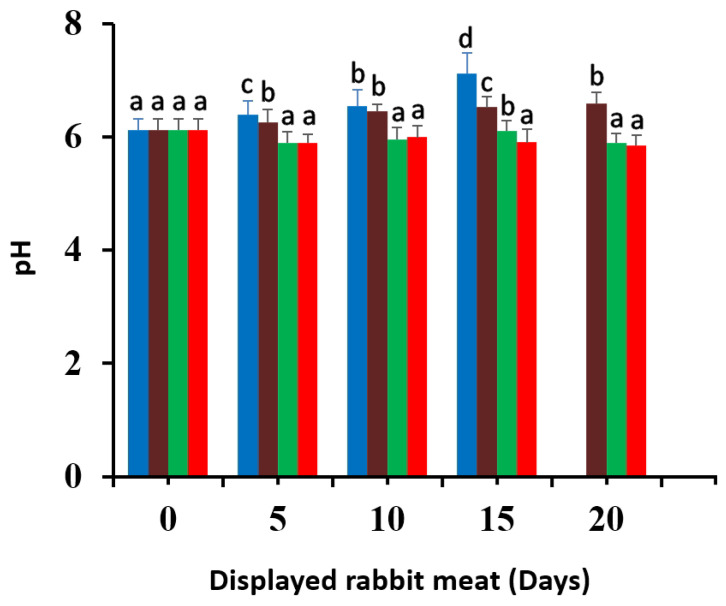
pH values of packed rabbit carcasses, displayed at 7 ± 1 °C for 20 days. Data represent means ± SD. (
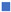
) 1-A unmarinated samples packed in air; (
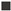
) 1-MA unmarinated samples packed in microaerophilic atmosphere; (
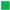
) 2-A marinated samples packed in air; (
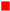
) 2-MA marinated samples packed in microaerophilic atmosphere. The lack of data (not determined) at 20 days of display for 1-A samples is due to excessive development of spoilage microorganisms and a noticeable unpleasant rancid off-odor. Different letters (a–d) indicate significant differences between treatments (*p* < 0.05).

**Figure 3 foods-12-02931-f003:**
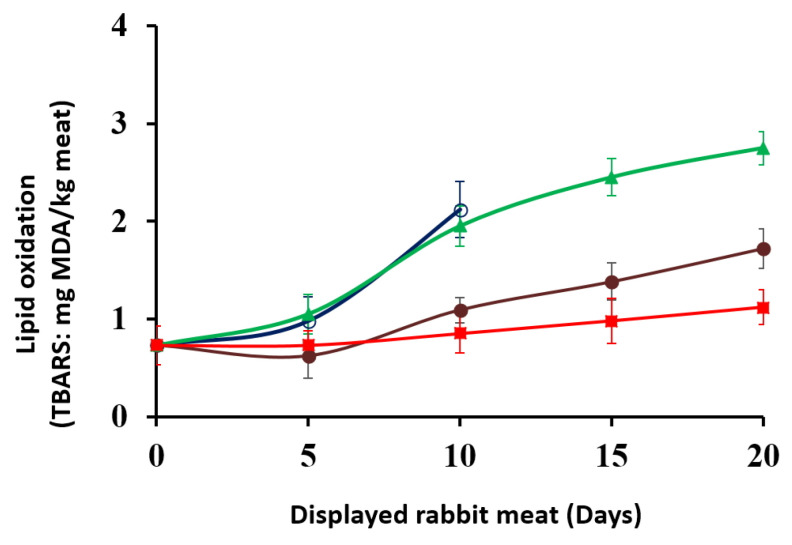
TBA-RS values of packed rabbit carcasses, displayed at 7 ± 1 °C for 20 days. Data represent means ± SD. (
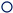
) 1-A unmarinated samples packed in air; (
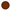
) 1-MA unmarinated samples packed in microaerophilic atmosphere; (
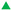
) 2-A marinated samples packed in air; (
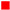
) 2-MA marinated samples packed in microaerophilic atmosphere. The lack of data (not determined) at an early stage (15 and 20 days) for 1-A samples is due to excessive development of spoilage microorganisms and a noticeable unpleasant rancid off-odor.

**Figure 4 foods-12-02931-f004:**
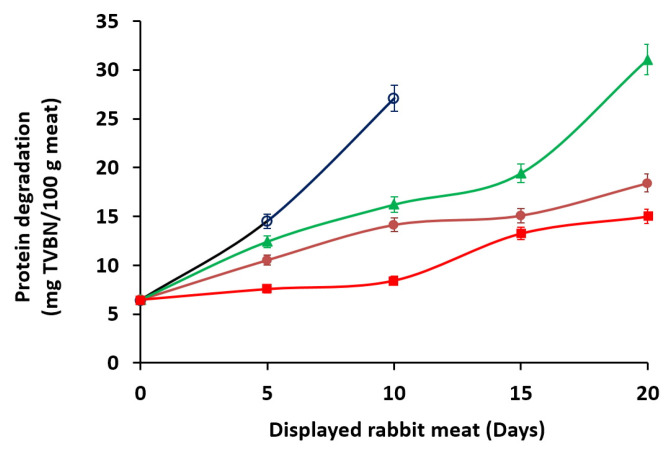
TVBN values of packed rabbit carcasses, displayed at 7 ± 1 °C for 20 days. Data represent means ± SD. (
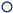
) 1-A unmarinated samples packed in air; (
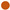
): 1-MA unmarinated samples packed in microaerophilic atmosphere; (
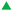
) 2-A marinated samples packed in air; (
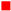
) 2-MA marinated samples packed in microaerophilic atmosphere. The lack of data (not determined) at an early stage (15 and 20 days) for 1-A samples is due to excessive development of spoilage microorganisms and a noticeable unpleasant rancid off-odor.

**Figure 5 foods-12-02931-f005:**
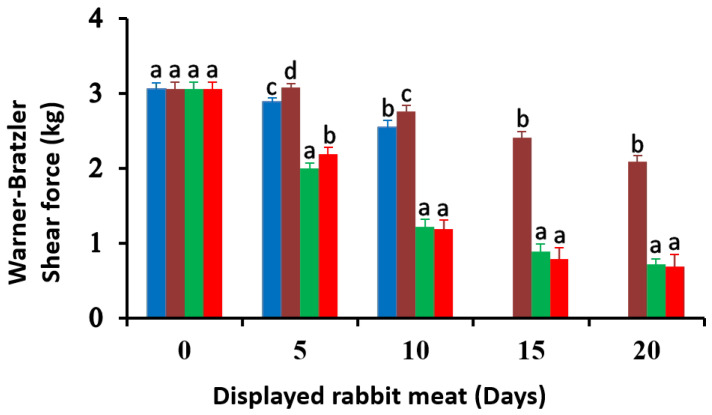
Shear force values of packed rabbit carcasses, displayed at 7 ± 1 °C for 20 days. Data represent means ± SD. (
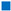
) 1-A unmarinated samples packed in air; (
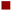
) 1-MA unmarinated samples packed in microaerophilic atmosphere; (
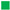
) 2-A marinated samples packed in air; (
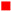
) 2-MA marinated samples packed in microaerophilic atmosphere. The lack of data (not determined) at an early stage (15 and 20 days) for 1-A samples is due to excessive development of spoilage microorganisms and a noticeable unpleasant rancid off-odor. Different letters (a–d) indicate significant differences between treatments (*p* < 0.05).

**Figure 6 foods-12-02931-f006:**
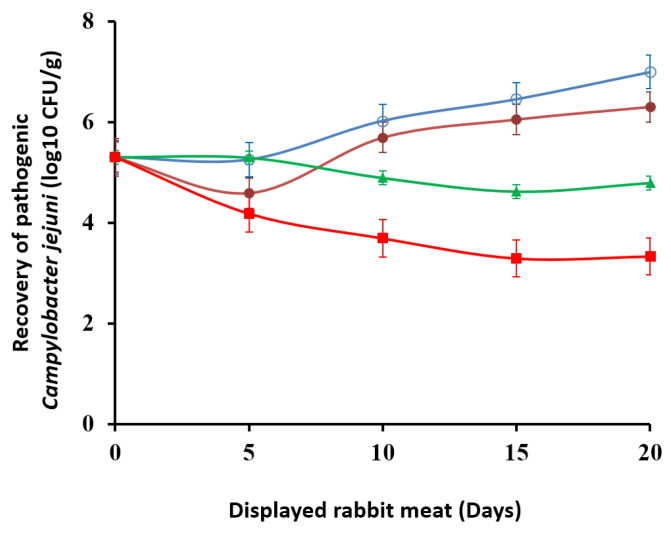
Survival of pathogen in packed rabbit carcasses contaminated with *C. jejuni* and displayed at 7 ± 1 °C for 20 days. Data represent means ± SD. (
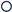
) 1-CA unmarinated samples packed in air; (
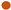
): 1-CMA unmarinated samples packed in microaerophilic atmosphere; (
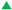
) 2-CA marinated samples packed in air; (
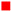
) 2-CMA marinated samples packed in microaerophilic atmosphere.

**Table 1 foods-12-02931-t001:** Average microbial counts (log_10_ CFU/g ± SD) values of packed rabbit carcasses displayed at 7 ± 1 °C for 20 days.

Displayed Rabbit Meat (Days)						
Microbial Group	Treatment ^1^	0	5	10	15	20
Psychrotrophic ^3,4^ bacteria	1-A	3.50 ± 0.02 ^av^	6.40 ± 0.01 ^cw^	7.59 ± 0.02 ^dx^	nd ^2^	nd
	1-MA	3.50 ± 0.01 ^av^	4.35 ± 0.03 ^bx^	4.64 ± 0.02 ^by^	5.90 ± 0.02 ^bcz^	6.89 ± 00.4 ^bw^
	2-A	3.50 ± 00.2 ^av^	4.25 ± 0.03 ^bvw^	6.39 ± 0.04 ^cx^	7.21 ± 0.04 ^dy^	nd
	2-MA	3.50 ± 0.01 ^av^	2.85 ± 0.02 ^av^	3.12 ± 0.03 ^av^	5.12 ± 0.02 ^aw^	4.89 ± 0.02 ^aw^
*Pseudomonas* ^3,4^ spp.	1-A	2.02 ± 0.01 ^av^	3.52 ± 0.02 ^bw^	4.59 ± 0.03 ^dx^	6.12 ± 0.04 ^dy^	7.89 ± 0.04 ^cz^
	1-MA	2.02 ± 0.01 ^aw^	1.10 ± 0.01 ^av^	2.15 ± 0.01 ^bw^	3.45 ± 0.02 ^bx^	3.56 ± 0.01 ^bx^
	2-A	2.02 ± 0.02 ^av^	3.25 ± 0.02 ^bw^	3.39 ± 0.02 ^cw^	4.25 ± 0.02 ^bcx^	3.85 ± 0.02 ^bwx^
	2-MA	2.02 ± 0.01 ^aw^	1.25 ± 0.01 ^av^	1.25 ± 0.01 ^av^	1.85 ±0.01 ^avw^	2.09 ± 0.01 ^aw^
*Brochothrix* *thermosphacta* ^3,4^	1-A	2.35 ± 0.02 ^av^	2.40 ± 0.01 ^bv^	3.24 ± 0.02 ^bw^	3.52 ±0.02 ^abw^	4.58 ± 0.02 ^bcx^
	1-MA	2.35 ± 0.01 ^av^	2.52 ± 0.01 ^bvw^	3.04 ± 0.01 ^bw^	3.74 ± 0.02 ^bx^	4.89 ±0.03 ^cy^
	2-A	2.35 ± 0.02 ^av^	2.89 ± 0.01 ^bvw^	4.39 ± 0.02 ^cx^	5.52 ± 0.03 ^cy^	4.18 ± 0.02 ^abx^
	2-MA	2.35 ± 0.01 ^aw^	1.29 ± 0.01 ^av^	2.02 ± 0.01 ^awx^	3.02 ± 0.02 ^ay^	3.72 ± 0.02 ^az^

Data represent means ± SD; ^1^ 1-A unmarinated packed in air, 1-MA unmarinated packed in low-O_2_ modified atmosphere, 2-A marinated packed in air, 2-MA marinated packed in low-O_2_ modified atmosphere; ^2^ not determined (due to excessive development of spoilage microorganisms and a noticeable unpleasant rancid off-odor); ^3^ Values followed by different letters (a–d) in a column are significantly different (*p* < 0.05); ^4^ Values followed by different letters (v–z) in a row are significantly different (*p* < 0.05).

**Table 2 foods-12-02931-t002:** Average lightness (L*), redness (a*), and yellowness (b*) values ^1,2^ of packed rabbit carcasses displayed at 7 ± 1 °C for 20 days.

Displayed Rabbit Meat (Days)						
Color attributes	Treatment ^3^	0	5	10	15	20
**Lightness (L*)**	1-A	62.12 ± 2.50 ^bw^	58.68 ± 2.16 ^av^	60.18 ± 2.08 ^bvx^	nd ^4^	nd
	1-MA	63.58 ± 3.09 ^bx^	57.45 ± 2.20 ^av^	58.79 ± 3.10 ^abw^	57.77 ± 2.10 ^bv^	58.89 ± 2.04 ^bw^
	2-A	58.35 ± 1.49 ^az^	56.98 ± 3.49 ^ay^	54.52 ± 2.39 ^ax^	51.28 ± 3.08 ^aw^	49.78 ± 1.52 ^av^
	2-MA	57.10 ± 3.52 ^ay^	57.96 ± 1.52 ^ay^	56.14 ± 1.89 ^ax^	52.88 ± 1.79 ^aw^	45.29 ± 1.86 ^bv^
**Redness (a*)**	1-A	3.37 ± 0.87 ^av^	4.89 ± 0.49 ^aw^	4.02 ± 0.39 ^aw^	nd	nd
	1-MA	3.58 ± 0.39 ^av^	5.12 ± 0.60 ^aw^	5.10 ± 0.49 ^aw^	4.16 ± 0.19 ^avw^	3.78 ± 0.33 ^av^
	2-A	5.47 ± 0.80 ^bv^	6.02 ± 1.09 ^bavw^	6.45 ± 0.52 ^bw^	5.56 ± 0.29 ^abv^	7.89 ± 0.72 ^bx^
	2-MA	5.19 ± 0.69 ^bv^	6.48 ± 0.79 ^bw^	6.25 ± 0.63 ^bw^	6.09 ± 0.35 ^bvw^	7.79 ± 0.41 ^bx^
**Yellowness (b*)**	1-A	6.45 ± 0.88 ^av^	7.87 ± 0.44 ^aw^	7.89 ± 0.43 ^bw^	nd	nd
	1-MA	6.10 ± 0.79 ^aw^	8.15 ± 0.30 ^abx^	6.17 ± 0.32 ^aw^	5.10 ± 0.67 ^av^	4.25 ± 0.31 ^av^
	2-A	8.39 ± 0.55 ^bv^	9.12 ± 0.58 ^bvw^	8.85 ± 0.37 ^cbvw^	8.99 ± 0.58 ^bvw^	9.41 ± 0.62 ^bw^
	2-MA	8.89 ± 0.42 ^bv^	8.59 ± 0.61 ^bv^	9.12 ± 0.44 ^cv^	8.97 ± 0.69 ^bv^	8.99 ± 0.60 ^bv^

Data represent means ± SD; ^1^ Values followed by different letters (a–c) in a column are significantly different (*p* < 0.05); ^2^ values followed by different letters (v–z) in a row are significantly different (*p* < 0.05); ^3^ 1-A unmarinated packed in air, 1-MA unmarinated packed in low-O_2_ modified atmosphere, 2-A marinated packed in air, 2-MA marinated packed in low-O_2_ modified atmosphere; ^4^ not determined due to excessive development of spoilage microorganisms and a noticeable unpleasant rancid off-odor.

**Table 3 foods-12-02931-t003:** Released water (%) ^1^ and cooking loss (%) ^2^ of packed rabbit carcasses and displayed at 7 ± 1 °C for 20 days. Data represent means ± SD.

Displayed Rabbit Meat (Days)					
Attributes	Treatment ^3^	5	10	15	20
Water holding capacity (RW%) ^1^	1-A	1.22 ± 0.08 ^av^	2.64 ± 0.29 ^bw^	Nd ^4^	nd
	1-MA	1.29 ± 0.06 ^av^	1.36 ± 0.05 ^av^	2.06 ± 0.13 ^bw^	2.39 ± 0.18 ^bw^
	2-A	1.29 ± 0.12 ^av^	1.32 ± 0.04 ^av^	1.31 ± 0.15 ^abv^	1.62 ± 0.05 ^av^
	2-MA	1.36 ± 0.05 ^av^	1.36 ± 0.05 ^av^	1.24 ± 0.22 ^av^	1.38 ± 0.01 ^av^
Cooking loss (%) ^2^	1-A	21.64 ± 1.28 ^bv^	25.55 ± 2.19 ^cw^	nd	nd
	1-MA	20.13 ± 1.79 ^bv^	21.34 ± 1.14 ^bv^	22.20 ± 0.95 ^bvw^	23.10 ± 0.94 ^bw^
	2-A	16.17 ± 1.53 ^aw^	14.17 ± 1.53 ^av^	17.84 ± 1.29 ^ax^	15.59 ± 2.72 ^aw^
	2-MA	14.11 ± 1.05 ^av^	15.25 ± 0.60 ^aw^	18.83 ± 2.62 ^ax^	14.92 ± 1.01 ^avw^

Data represent means ± SD. ^1^ Values followed by different letters (a–c) in a column are significantly different (*p* < 0.05); ^2^ values followed by different letters (v–x) in a row are significantly different (*p* < 0.05); ^3^ 1-A unmarinated packed in air, 1-MA unmarinated packed in low-O_2_ modified atmosphere, 2-A marinated packed in air, 2-MA marinated packed in low-O_2_ modified atmosphere; ^4^ not determined (due to excessive development of spoilage microorganisms and a noticeable unpleasant rancid off-odor).

**Table 4 foods-12-02931-t004:** Sensory attributes and acceptability ^1,2^ of packed rabbit carcasses displayed at 7 ± 1 °C for 20 days. Data represent means ± SD.

Displayed Rabbit Meat (Days)						
Attributes	Treatment ^3^	0	5	10	15	20
Spicy intensity ^4^	1-A	1.00 ± 0.00 ^av^	1.00 ± 0.00 ^av^	1.00 ± 0.00 ^av^	Nd ^6^	nd
	1-MA	1.00 ± 0.00 ^av^	1.00 ± 0.00 ^av^	1.00 ± 0.00 ^av^	1.00 ± 0.00 ^av^	1.00 ± 0.00 ^av^
	2-A	2.67 ± 0.49 ^bx^	2.33 ± 0.49 ^bw^	2.17 ± 0.39 ^bvw^	1.83 ± 0.39 ^bv^	1.50 ± 0.52 ^bv^
	2-MA	2.50 ± 0.52 ^bx^	2.50 ± 0.52 ^bx^	2.00 ± 0.00 ^bvw^	1.67 ± 0.49 ^abv^	1.50 ± 0.52 ^bv^
Odor intensity ^4^	1-A	1.00 ± 0.00 ^av^	1.33 ± 0.49 ^av^	3.17 ± 0.39 ^bw^	nd	nd
	1-MA	1.17 ± 0.39 ^av^	1.00 ± 0.00 ^av^	1.33 ± 0.49 ^av^	1.83 ± 0.39 ^aw^	2.33 ± 0.49 ^ax^
	2-A	1.00 ± 0.00 ^av^	1.17 ± 0.39 ^av^	1.50 ± 0.52 ^avw^	2.83 ± 0.39 ^bw^	3.17 ± 0.39 ^bx^
	2-MA	1.17 ± 0.39 ^av^	1.00 ± 0.00 ^av^	1.17 ± 0.39 ^av^	1.67 ± 0.49 ^avw^	2.17 ± 0.39 ^aw^
Overall acceptability ^5^	1-A	5.00 ± 0.00 ^aw^	4.67 ± 0.49 ^aw^	2.67 ± 0.49 ^av^	nd	nd
	1-MA	5.00 ± 0.00 ^aw^	4.83 ± 0.39 ^aw^	3.17 ± 0.39 ^bav^	3.00 ± 0.60 ^av^	2.83 ± 0.39 ^av^
	2-A	5.00 ± 0.00 ^aw^	4.50 ± 0.52 ^aw^	2.83 ± 0.39 ^av^	2.67 ± 0.49 ^av^	2.50 ± 0.52 ^av^
	2-MA	5.00 ± 0.00 ^aw^	5.00 ± 0.00 ^av^	4.50 ± 0.52 ^cv^	4.33 ± 0.49 ^bv^	4.83 ± 0.39 ^bv^

Data represent means ± SD; ^1^ values followed by different letters (a–c) in a column are significantly different (*p* < 0.05); ^2^ values followed by different letters (v–x) in a row are significantly different (*p* < 0.05); ^3^ 1-A unmarinated packed in air, 1-MA unmarinated packed in low-O_2_ modified atmosphere, 2-A marinated packed in air, 2-MA marinated packed in low-O_2_ modified atmosphere; ^4^ spicy and odor intensity were scored on a 5-point scale (1 = none, 5 = extreme); ^5^ overall acceptability was scored on a 5-point scale (1 = not acceptable, 5 = highly acceptable); ^6^ not determined (due to excessive development of spoilage microorganisms and a noticeable unpleasant rancid off-odor).

**Table 5 foods-12-02931-t005:** Effect of marinade treatment on the shelf life prolongation (days) of packed rabbit carcasses displayed at 7 ± 1 °C for 20 days. Data represent means ± SD.

Treatment ^1^	Shelf Life (Days)	Shelf Life Extension (%)
1-A	5	-
1-MA	>20	>300
2-A	10	100
2-MA	>20	>300

^1^ 1-A unmarinated packed in air, 1-MA unmarinated packed in low-O_2_ modified atmosphere, 2-A marinated packed in air, 2-MA marinated packed in low-O_2_ modified atmosphere.

## Data Availability

The data presented in this study are available in the article.

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
