# Peer review of "Use of Algerian Type Ras El-Hanout Spices Mixture with Marination to Increase the Sensorial Quality, Shelf Life, and Safety of Whole Rabbit Carcasses under Low-O2 Modified Atmosphere Packaging"

_foods, 2023, doi:10.3390/foods12152931_

Round 1

Reviewer 1 Report

The manuscript studied the effect of combined treatments with Ras El-Hanout spices mixture and marinade solution on sensorial quality, shelf life and safety of whole rabbit carcasses under low-O2 modified atmosphere packaging (MAP). These methods including marination and MAP have been widely used for the preservation of agricultural products. The innovation of this manuscript is weak. Because of some problems this manuscript requires revision.

1. In Results and Discussion section, the statistical analysis is missing, such as all Tables, Figure 1, and Figure 4. Different letters representing significant difference (P < 0.05) should be labeled in Tables or Figures.

2. In Results and Discussion section, the authors should compare the relevant papers with your study in order to demonstrate how your study is different from those that have already been published instead of similarity, and how your paper can fill the knowledge gap.

Author Response

Author's Reply to the Review Report (Reviewer #1)

Reviewer #1: The manuscript studied the effect of combined treatments with Ras El-Hanout spices mixture and marinade solution on sensorial quality, shelf life and safety of whole rabbit carcasses under low-O2 modified atmosphere packaging (MAP). These methods including marination and MAP have been widely used for the preservation of agricultural products. The innovation of this manuscript is weak. Because of some problems this manuscript requires revision.

Authors' response. We do sincerely thank this reviewer for their helpful and constructive comments. We have thoroughly reviewed the manuscript according to this reviewer's comments. A detailed response (comment by comment) can be found below.

Reviewer #1: In Results and Discussion section, the statistical analysis is missing, such as all Tables, Figure 1, and Figure 4. Different letters representing significant difference (P < 0.05) should be labeled in Tables or Figures.

Authors' response. We greatly appreciate your thoughtful comments that will undoubtedly contribute to improving the manuscript. We hope that all of your comments will be addressed accordingly in the present revised manuscript. Taking into account reviewer comments, the manuscript has been improved.

Reviewer #1: In Results and Discussion section, the authors should compare the relevant papers with your study in order to demonstrate how your study is different from those that have already been published instead of similarity, and how your paper can fill the knowledge gap.

Authors' response. Thank you for your comment. There is no previous research on Ras El-Hanout in whole rabbit packed in low O2 and Campylobacter jejuni atmospheres. However, and to enrich the discussion, we have tried to compare our results with other biopreservation methods. However, comparison with other published data is complicated, but comparison is still possible. Our results should be compared with those obtained with other plant extracts and related compounds, of similar or different origin, both under the same and different experimental conditions. We believe that this should be the focus of the discussion. Most researchers in different regions of the world adapt experimental methods to suit the possible applications in their particular field. In the original manuscript, what is novel compared to previous publications is clearly indicated so that comparison of results may suggest the existence of a correlation. In our opinion the approach to the discussion that has been adopted in our work is correct, the results and discussion carried out are always useful.

Regarding the point “How our paper can fill the knowledge gap”. The manuscript provides new scientific knowledge, and does not merely confirm what already exists in the scientific literature. Thus, the results are fit for comparison. Most of the laboratory work focused on the powerful antioxidant and antimicrobial activities of the composition of plant extracts (called chemotypes) and concluded that they may differ according to harvesting seasons, geographical sources, harvesting during or immediately after flowering, fresh and dried plant organ, genetic considerations of the plant, etc. Thus, it is possible that variation in composition among EOs is sufficient to cause variability in the degree of biological activity. To meet the challenge, the food industry has found in plant extracts a good alternative to replace synthetic products (for regulatory reasons and for consumers' reticence towards synthetic ingredients).

The antimicrobial and antioxidant mechanism should be similar, especially when dealing with a range of animal products (meat, fish, egg, milk...) and therefore, according to several authors, the results could be comparable. A large number of articles dealing with the antioxidant or antibacterial activity of herbal extracts for food preservation have used the same approach for the discussion of the results. Our conclusion is elaborated, sufficiently concise and concentrates mainly on the present findings.

Thank you very much for your comments and suggestions.

Yours sincerely,

Reviewer 2 Report

This is a very interesting article. Development of plant extracts with application to meat products have been tested for a long time in China. Therefore, it is good to see an interest from another geographical location.

Rabbit meat has unfortunately low appeal in, for example, Northern Europe because of its association with domestic pets. Evan in France, where rabbit meat can be more available than in other countries, its presence is also low. 

Notheless, despite you mentioning meat functionality, rabbit meat being lean should also have an appeal to those on diets (but then we go back to the perception of "domestic pet" status that rabbit meat has).

It would be interesting in the future to read on the results based on other stressors (heat, thawing) as well as other micro-organisms.

Author Response

Author's Reply to the Review Report (Reviewer #2)

Comments and Suggestions for Authors

Reviewer #2: This is a very interesting article. Development of plant extracts with application to meat products have been tested for a long time in China. Therefore, it is good to see an interest from another geographical location.

Authors' response. Thank you very much. We do sincerely thank this reviewer for their helpful and constructive comments.

Reviewer #2: Rabbit meat has unfortunately low appeal in, for example, Northern Europe because of its association with domestic pets. Evan in France, where rabbit meat can be more available than in other countries, its presence is also low.

However, despite you mentioning meat functionality, rabbit meat being lean should also have an appeal to those on diets (but then we go back to the perception of "domestic pet" status that rabbit meat has).

It would be interesting in the future to read on the results based on other stressors (heat, thawing) as well as other microorganisms.

Authors' response. Thank you very much for your suggestions and we have taken note for possible future works.

Thank you very much for your comments and suggestions.

Yours sincerely,

Reviewer 3 Report

Dear Authors, the article is very interesting and current. I think that not only rabbit carcasses can be treated in this way, but also other types of raw meat.
However, I was puzzled by a few issues, which I will recommend explaining later in the review.

My remarks start from the chapter "Materials and methods":
- do the Authors believe that marinating the whole rabbit carcass (not individual elements or half carcasses) is appropriate? It's all about the accuracy of mixing the marinade with the meat.

- I believe that the composition of the marinade should be given in % (g/100 g of the ingredient). The share given by the Authors (line 131 - 4:2:1:1) is not correct. The recipes are more based on percentages.

- where did the value of the 8% marinade addition (line 179) come from? Did the authors assume so or did they take into account their previous experience. It could be, for example, 10% or 3%? Why it is 8%?

- I can't quite understand why Campylobacter was chosen for the microbial contamination (Section 2.3)? This microflora is not characteristic of rabbit meat, except for poultry, which was visible in "Graphical abstract", in one of the photos.

- the title of subchapter 2.2 is "Optimization of the Spices Mixture and Marinade Solution". Why is there a fragment about the determination of TPCs (Total phenolic compounds) in this fragment? In addition, the content does not indicate the results of this determination in any table or graph.

- the time elapsed from the moment of slaughtering the rabbits to the moment of adding the marinade and further - to the determination was not indicated? Were these days (0, 5, 10, 15 and 20) counted from the addition of the marinade?

- the method of mixing the marinade (lines 180-181) requires explanation: how was it done, with a spoon or in some device?

- in line 185 the authors write about "manually homogenized". How was this done and what was it like for a rabbit carcass?

- line 193: was the experiment repeated in e.g. 3 series? Were these 3 replicates done on the same batch of slaughtered animals? The description is not clear to the reader.

- line 195: in how many replicates was the pH measured? Because with the color it was indicated that there were 10 measurements. Please complete this information.

- line 198: it is a bit difficult to imagine measuring the surface color parameters of meat to which sweet red pepper and turmeric have been added. Do you think that it could have had a big impact on the measured parameters?

- I have trouble with understanding the information given in lines 268-271 and 686-688 too: "A score value > 3 of any attribute (odor, spicy and color), denoted that rabbit meat was not acceptable by panelists. On the contrary, for overall acceptability, a score value < 3 denoted that rabbit meat was not acceptable by panelists". I have the impression that they contradict each other, so I am asking for a different record of this information. Because I don't know what value was acceptable at the moment. And how (table 4) e.g. with "spicy intensity" rated at 1, "odor intensity" also rated at 1, it came out that "overall acceptability" was rated at 5?

- tables: I suggest that information such as "Data represent means ± SD" should be placed under the table, when describing variants.

- line 360: from the technologist's point of view, "... fresh meat" is not the same as "marinated meats", and the context shows that they are the same group of products.

- line 389: The Authors write regarding their color measurement results: "A similar phenomenon was reported for marinated pork loin slices vacuum-packaged by addition of oil/lemon/beer with the inclusion of juniper, oregano and rosemary essential oil [ 41]". We can not compare the composition of a marinade that does not contain any coloring spices with one that contains at least 2 coloring ingredients.

- I believe that the explanation of the results in lines 415-416 and the same for the TVBN results with the results of microbiological determinations is incorrect. After all, the results of microbiology are described only from line 591, which is much later. This is probably a premature conclusion.

- why the Authors wrote chapter "3.9 Shelf-life". It looks like an article summary, and it really is. I believe that this information is a repetition of the conclusions from individual subsections and should not be included here.

- References: please carefully review the record of individual literature items to ensure that they are consistent with the requirements of the journal.

Author Response

Author's Reply to the Review Report (Reviewer #3)

We do sincerely thank this reviewer for their helpful and constructive comments. Thank you very much for your effort. We have thoroughly reviewed the manuscript according to this reviewer's comments. A detailed response (comment by comment) can be found below.

Reviewer #3: The article is very interesting and current. I think that not only rabbit carcasses can be treated in this way, but also other types of raw meat.

Authors' response. Thank you very much for your comments. The reviewer is right. We are well aware of this fact! Just look at the number of articles mentioned in your manuscript to really know the wide variety that exists in the scientific literature on marinating raw meat.

Reviewer #3: do the Authors believe that marinating the whole rabbit carcass (not individual elements or half carcasses) is appropriate? It's all about the accuracy of mixing the marinade with the meat.

Authors' response. Yes of course! In the Algerian market, rabbit meat is often sold as whole carcasses and several companies currently prefer to sell eviscerated whole rabbit carcasses, as it saves processing time and cut-up labor and machine costs. However, weight loss due to cut up leads to economic losses for rabbit processors. As rabbit products are sold by weight, any weight loss is economically undesirable. The reviewer is right; it is all about the accuracy of mixing the marinade with the meat. Many other arguments are well explained in the text

Reviewer #3:  I believe that the composition of the marinade should be given in % (g/100 g of the ingredient). The share given by the Authors (line 131 - 4:2:1:1) is not correct. The recipes are more based on percentages.

Response: Thanks for your comment. We improved the paragraph based on your suggestions. Appropriate expression of marinade composition has been corrected in the revised manuscript. The expression in the text becomes "Based on the preliminary results, the final marinade solution selected for the main experiment was composed of EVOO/lemon/garlic/onion (4/2/1/1, In other words: 50/25/12.5/12.5%) with a Ras El-Hanout mixture at 500 mg/kg (0.05%).

Reviewer #3:  where did the value of the 8% marinade addition (line 179) come from? Did the authors assume so or did they take into account their previous experience. It could be, for example, 10% or 3%? Why it is 8%?

Authors' response. We thank you very much for the comments and suggestions. The comments and suggestions are valuable and very helpful for revising and improving our manuscript. According to reviewer comments, section of the paragraph “2.4. Rabbit Carcasses Treatments” has been re-written in order to include more details about the experimental conditions carried out. We stated in the Materials and Methods section (new version) that 8% is the uptake value of marinade (marination retention); indicating that the additive included in the marinade solution increased carcass water absorption. Immediately after marinating, each marinated carcass was drained for 30 min and weighed again. Marinade uptake was calculated based on carcass weight of before marination (W1) and its weight after marination (W2), according to the equation:

Marinade uptake (%) = [(W2 –W1)/W1] × 100

The text has been modified

Reviewer #3:  I can't quite understand why Campylobacter was chosen for the microbial contamination (Section 2.3)? This microflora is not characteristic of rabbit meat, except for poultry, which was visible in "Graphical abstract", in one of the photos.

Authors' response. The reviewer is absolutely right! In recent years, surveillance programs in some developed countries have revealed a high level of Campylobacter contamination in poultry and poultry products. According to the European Food Safety Authority (EFSA), up to 80% of human campylobacterioses are attributed to "poultry" as a reservoir. To justify our choice, in Algeria, it is common for chickens and rabbits to be raised together with the consequent risk of Campylobacter transmission between the two species.

Reviewer #3:  the title of subchapter 2.2 is "Optimization of the Spices Mixture and Marinade Solution". Why is there a fragment about the determination of TPCs (Total phenolic compounds) in this fragment? In addition, the content does not indicate the results of this determination in any table or graph.

Authors' response. This is a very good indicator of the biological activity of bioactive compounds such as polyphenols that the marinade solution may contain! The result of this determination is well explained in the text (cf. section 3.10. Challenge trial with Campylobacter jejuni).

It is written: Lines 814-819

“As already mentioned, the final marinade solution contained large amounts of polyphenols (535.29 mg GAE/g), which are already recognized for their antimicrobial activity. These results were consistent with those of our previous works in which olive leaves extract from different varieties of olive trees showed antimicrobial effects in turkey meat, raw minced beef and camel meat during display. Olive leaves extract contains great quantities of polyphenols (216.5 mg GAE/g)”.

Reviewer #3:  the time elapsed from the moment of slaughtering the rabbits to the moment of adding the marinade and further - to the determination was not indicated? Were these days (0, 5, 10, 15 and 20) counted from the addition of the marinade?

Authors' response. All rabbit carcasses were aseptically transported cold (1 ± 1 °C) to the laboratory within 2 hours post-mortem. On the day after slaughter (6 h post-mortem), eviscerated rabbit carcasses were individually weighed and assigned to one of the two group treatments (32 carcasses per group):

Group 1: Unmarinated rabbit carcasses (n = 32)

Group 2: Marinated rabbit carcasses (n = 32)

Reviewer #3: Were these days (0, 5, 10, 15 and 20) counted from the addition of the marinade?

Authors' response. Yes of course! 0, 5, 15, and 20 days represent retail display selected sampling time for all samples. 20 days is the total retail-display period for all the samples (Unmarinated rabbit carcasses and those marinated).

Reviewer #3:  the method of mixing the marinade (lines 180-181) requires explanation: how was it done, with a spoon or in some device?

Authors' response. Thank you for your helpful comment. To ensure better marinating operation, the marinating solution was thoroughly mixed for 1 minute using a laboratory blender. The carcass samples were treated with a brush designed for marinades.

Reviewer #3:  in line 185 the authors write about "manually homogenized". How was this done and what was it like for a rabbit carcass?

Authors' response. It's confusion on our part! Based on reviewers' comments, the sentence has been rewritten to include more detail about the experimental conditions performed. “In order to ensure a good distribution of pathogenic microorganism over the whole surface rabbit carcasses, the latter were manually stirred for 1 min, which represents the time needed to obtain a good surface distribution”.

Reviewer #3:  line 193: was the experiment repeated in e.g. 3 series? Were these 3 replicates done on the same batch of slaughtered animals? The description is not clear to the reader.

Authors' response. Thanks for the comment. These 3 replicates were done on the same batch of slaughtered rabbits. An independent repeated analysis of each attribute is usually performed, followed by calculating the mean and the standard deviation. The goal is to observe the expected variation in a test result under normal laboratory operating conditions.

Reviewer #3:  line 195: in how many replicates was the pH measured? Because with the color it was indicated that there were 10 measurements. Please complete this information.

Authors' response. As explained above, 3 experiment replicates were done on the same batch of slaughtered rabbits and an independent repeated analysis of each attribute is usually performed, followed by calculating the mean and the standard deviation. For the pH, 3 measurements were determined from each replicate (9 measurements in total). Color measurement makes it possible to characterize a color according to 3 parameters: The lightness ( L*), The redness (a*) and the yellowness (b*). In general, the color meat surface should be measured on several points (> 10) due to the meat surface heterogeneity (meat sample geometrical form, state of the fibers, surface meat pigment, blood and fat on the surface etc,). Thus, 10 measurements were determined from each replicate (30 measurements in total).

Reviewer #3:  line 198: it is a bit difficult to imagine measuring the surface color parameters of meat to which sweet red pepper and turmeric have been added. Do you think that it could have had a big impact on the measured parameters?

Authors' response. Yes of course! The effect of sweet red pepper and turmeric could have a strong influence on the measured color surface, especially during the first days of storage! The main goal of this work was assessing the preservative potential of Ras El-Hanout marinade solution based on its antimicrobial/antioxidant properties, to extend rabbit meat shelf life at concentrations that enable a balance between the sensory acceptability and its biological efficacy. For this purpose, it must be taken into account Ras El-Hanout marinade solution impact on the color properties of rabbit meat to exploit its preservative and flavoring properties.

A comment on this possibility was already included in the text.

Reviewer #3:  I have trouble with understanding the information given in lines 268-271 and 686-688 too: "A score value > 3 of any attribute (odor, spicy and color), denoted that rabbit meat was not acceptable by panelists. On the contrary, for overall acceptability, a score value < 3 denoted that rabbit meat was not acceptable by panelists". I have the impression that they contradict each other, so I am asking for a different record of this information. Because I don't know what value was acceptable at the moment. And how (table 4) e.g. with "spicy intensity" rated at 1, "odor intensity" also rated at 1, it came out that "overall acceptability" was rated at 5?

Authors' response. Thank you for your comment because it helped us to realize that we had expressed it wrong. We apologize for the error that occurred! In fact, it should have read "A score value > 3 of any attribute (odor and spicy)” instead "A score value > 3 of any attribute (odor, spicy and color)”.

In the text, in order to avoid similarities, we have indicated the work reference on which we relied. For descriptive test, panelists evaluated for each sample. The attributes ‘spicy’ and ‘odor’ intensities were rated using a 5-point descriptive scale, using a paper scorecard. Scores for ‘odor’ referred to the intensity of off odors associated to meat spoilage: 1 = none; 2 = slight; 3 = small; 4 =moderate; and 5 = extreme. The attribute ″ spicy odor″ referred to the intensity of perceptible spicy odor: 1 = none; 2 = slight; 3 = small; 4 = moderate; and 5 = extreme. On the other hand, a common way to assess overall acceptability is through hedonic scales where the participants indicate how much they like or dislike the sample in terms of a specific sensory property. A score value higher than 3 denoted that rabbit meat was not acceptable by panelists probably due to high spicy intensity. It is not a contradiction as mentioned by the reviewer but a different expression of two different sensory paradigms (descriptive and hedonic).

You can read on the footnote legend (sensory Table 4):

“Spicy and odor intensity were scored on a 5-point scale (1= none, 5 = extreme); Overall acceptability was scored on a 5-point scale (1 = not acceptable, 5 = highly acceptable)”.

Reviewer #3:  tables: I suggest that information such as "Data represent means ± SD" should be placed under the table, when describing variants.

Authors' response. Thank you very much for your comments that helped us improve this manuscript.

Reviewer #3:  line 360: from the technologist's point of view, "... fresh meat" is not the same as "marinated meats", and the context shows that they are the same group of products.

Authors' response. The reviewer is right. However, both types of meat are considered raw! And, therefore, they are subject to intense alterations during storage.

Reviewer #3:  line 389: The Authors write regarding their color measurement results: "A similar phenomenon was reported for marinated pork loin slices vacuum-packaged by addition of oil/lemon/beer with the inclusion of juniper, oregano and rosemary essential oil [ 41]". We can not compare the composition of a marinade that does not contain any coloring spices with one that contains at least 2 coloring ingredients.

Authors' response. Thanks for your comment. There is no previous investigation about Ras El-Hanout on rabbit and pathogenic bacteria. However, we compared our results to other extracts. It is for this reason that the comparison of the published data is complicated but which remains comparable. Our results should be compared to those of other marinade solution and related compounds (especially phenolic compounds) from the same and different origin under the same and different conditions. This is what the discussion should focus on.

Reviewer #3:  I believe that the explanation of the results in lines 415-416 and the same for the TVBN results with the results of microbiological determinations is incorrect. After all, the results of microbiology are described only from line 591, which is much later. This is probably a premature conclusion.

Authors' response. We deeply appreciate reviewer comments. Authors are thankful to the reviewer for their kind appreciation and valuable suggestions to improve the quality of manuscript. Following your comment, we have decided to transfer the "microbiological analysis" part just before the "color evaluation" part. In addition, this way we avoid premature conclusions

Reviewer #3:  why the Authors wrote chapter "3.9 Shelf-life". It looks like an article summary, and it really is. I believe that this information is a repetition of the conclusions from individual subsections and should not be included here.

Authors' response. Thanks for your comment. The main goal of this work was assessing the preservative potential of Ras El-Hanout marinated solution based on its antioxidant /antimicrobial properties, to extend marinated rabbit meat shelf-life at concentrations that enable a balance between the sensory acceptability and its biological efficacy. After all, the results of shelf life are described much later. We believe that discussion details from the shelf life study should be provided. This is what the discussion should focus on. Moreover, this aspect puts our results in potential interest on the part of the rabbit sector! I must say that our study it is not a predictive study using prediction models but rather a simple estimate of the marinated rabbit meat additional shelf life following the application of a bio preservative. We were inspired by microbiological, chemical and sensory critical thresholds already established according to the scientific literature. We have defined the shelf life as “the period between processed marinated rabbit meat and the sampling, during which the product is in a state of satisfactory quality in terms of chemical, microbiological and sensorial”. All these information’s are already indicated in the text in order to enrich the discussion.

Reviewer #3:  References: please carefully review the record of individual literature items to ensure that they are consistent with the requirements of the journal.

Authors' response. Thanks for your comment. Errors have been corrected

Thank you very much for your comments and suggestions.

Yours sincerely,

Reviewer 4 Report

In this study, authors examined the effect of Ras El-Hanout Spices marination treatment on the rabbit meat quality with or without MAP packaging.

The experiment is very controlled, however, the manuscript should be revised taking into account the below criticisms.

Major

1. The length of article is too long

The fisrt impression was discussions are too long for the performed experiments. Most readers must want to know the results first, however, each long discussion disturbs to reach the results. I highly recommend to separate Results and Discussion part, and authors should consider writing the discussion very concisely. I could not count the words correctly, but they appear to exceed 10,000 words. I hope authors to summarize the content 6,000 words at most.

Minor

1. Line 280-285

These sentences should be placed in the Materials and Methods (after the line 111-115). These are not results.

2. The lack of data

Authors must explain the reasons lacking of 15-day and 20-day of 1-A at early stage, maybe in the methods section. It was very confusing these data are not shown in many figures. The reason was shown in only in Tables.

Furthermore, authors described a few data of 15 days and 20 days of 1-A; Figure 1 (15 days); TBARS values of 20 days (line 335). This is very odd.

3. line487

3th -> 3rd

4. line542

WRC -> WHC

Most English was fine. The errors were indicated in minor issues.

Author Response

Author's Reply to the Review Report (Reviewer #4)

In this study, authors examined the effect of Ras El-Hanout Spices marination treatment on the rabbit meat quality with or without MAP packaging.

The experiment is very controlled, however, the manuscript should be revised taking into account the below criticisms.

Authors' response. We do sincerely thank this reviewer for their helpful and constructive comments. We have thoroughly reviewed the manuscript according to this reviewer's comments. A detailed response (comment by comment) can be found below.

Major

  1. The length of article is too long

The fisrt impression was discussions are too long for the performed experiments. Most readers must want to know the results first, however, each long discussion disturbs to reach the results. I highly recommend to separate Results and Discussion part, and authors should consider writing the discussion very concisely. I could not count the words correctly, but they appear to exceed 10,000 words. I hope authors to summarize the content 6,000 words at most.

Authors' response. We are well aware of the length of our article! However, given the volume of results obtained, we thought, on the one hand, to describe our methodology well and on the other hand, to enrich our discussion.

“Foods has no restrictions on the maximum length of manuscripts. Full experimental details must be provided so that the results can be reproduced. Foods requires that authors publish all experimental controls and make full datasets available where possible”. We thank you in advance for your understanding.

Minor

Reviewer #4: Line 280-285: These sentences should be placed in the Materials and Methods (after the line 111-115). These are not results.

Authors' response. Thank you for your helpful comment. Following your comment, we checked and corrected the paragraph.

Reviewer #4: The lack of data. Authors must explain the reasons lacking of 15-day and 20-day of 1-A at early stage, maybe in the methods section. It was very confusing these data are not shown in many figures. The reason was shown in only in Tables.

Authors' response. It’s corrected. Thank you very much.

Reviewer #4: Furthermore, authors described a few data of 15 days and 20 days of 1-A; Figure 1 (15 days); TBARS values of 20 days (line 335). This is very odd.

Authors' response. This is our mistake, sorry! We determined this value on the 20th day only for TBA-RS and we decided not to do it with the other attributes due to excessive development of spoilage microorganisms and rancid off-odor. The text has been modified.

Reviewer #4: 3. line 487: 3th -> 3rd

Authors' response. It is corrected. Thank you very much.

Reviewer #4: 4. line 542: WRC -> WHC

Authors' response. It is corrected. Thank you very much.

Thank you very much for your comments and suggestions.

Yours sincerely,

Round 2

Reviewer 1 Report

Because of some problems this manuscript requires revision.

1. In Figure 1 and Figure 4, the statistical analysis is missing.

2. In Table 1, Table 2, Table 3, and Table 4, “values followed by different letters (a-d) in a column are significantly different (p < 0.05)”, did the authors mark a, b, c, and d from highest to lowest for the order of values? There is an error in the different letters for statistical analysis. Please carefully check. In addition, “Values followed by different letters (w-z) in a row are significantly different (p < 0.05)”, what does letter “v”mean? It should be marked with letters starting from v?

Author Response

Author's Reply to the Review Report (Reviewer #1)

Reviewer #1:  In Figure 1 and Figure 4, the statistical analysis is missing.

Authors' response. This is our mistake, sorry! It is corrected. Thank you very much.

Reviewer #1:  In Table 1, Table 2, Table 3, and Table 4, “values followed by different letters (a-d) in a column are significantly different (< 0.05)”, did the authors mark a, b, c, and d from highest to lowest for the order of values? There is an error in the different letters for statistical analysis. Please carefully check. In addition, “Values followed by different letters (w-z) in a row are significantly different (< 0.05)”, what does letter “v”mean? It should be marked with letters starting from v?

Authors' response. We do sincerely thank this reviewer for their helpful and constructive comments. We have thoroughly reviewed the manuscript according to this reviewer's comments.

Reviewer #1:  did the authors mark a, b, c, and d from highest to lowest for the order of values?

Authors' response. We marked a, b, c and d from lowest to highest for the order of values.

Reviewer #1:  what does letter “v”mean? It should be marked with letters starting from v? Authors' response. This is our mistake, sorry!  It is corrected. Thank you very much.

Thank you very much for your comments and suggestions.

Yours sincerely,

Reviewer 4 Report

Foods has no restrictions on the maximum length of manuscripts, provided that the text is concise and comprehensive.

I pointed that the manuscript was not concise, especially the discussion part was redundant. To describe the methodology well and on the other hand, to brief the discussion is possible and very important in terms of writing a scientific paper.

Anyway, authors revised the manuscript according to the suggestions. I would like the editor to judge if the redundancy of the article is acceptable.

Author Response

Author's Reply to the Review Report (Reviewer #4)

Reviewer #4:  Foods has no restrictions on the maximum length of manuscripts, provided that the text is concise and comprehensive. I pointed that the manuscript was not concise, especially the discussion part was redundant. To describe the methodology well and on the other hand, to brief the discussion is possible and very important in terms of writing a scientific paper.

Anyway, authors revised the manuscript according to the suggestions. I would like the editor to judge if the redundancy of the article is acceptable.

Authors' response. Thank you very much. We do sincerely thank this reviewer for their helpful and constructive comments.

Thank you very much for your comments and suggestions.

Yours sincerely,
